# Activation pathway of a G protein-coupled receptor uncovers conformational intermediates as targets for allosteric drug design

Shaoyong Lu [1,2,9✉], Xinheng He [3,4,9], Zhao Yang[5,9], Zongtao Chai[6,9], Shuhua Zhou[5,9], Junyan Wang[5], Ashfaq Ur Rehman[2], Duan Ni [2], Jun Pu[7], Jinpeng Sun [5✉] & Jian Zhang [1,2,8✉]

G protein-coupled receptors (GPCRs) are the most common proteins targeted by approved drugs. A complete mechanistic elucidation of large-scale conformational transitions underlying the activation mechanisms of GPCRs is of critical importance for therapeutic drug development. Here, we apply a combined computational and experimental framework integrating extensive molecular dynamics simulations, Markov state models, site-directed mutagenesis, and conformational biosensors to investigate the conformational landscape of the angiotensin II (AngII) type 1 receptor ($AT_1$ receptor) — a prototypical class A GPCR— activation. Our findings suggest a synergistic transition mechanism for $AT_1$ receptor activation. A key intermediate state is identified in the activation pathway, which possesses a cryptic binding site within the intracellular region of the receptor. Mutation of this cryptic site prevents activation of the downstream G protein signaling and β-arrestin-mediated pathways by the endogenous AngII octapeptide agonist, suggesting an allosteric regulatory mechanism. Together, these findings provide a deeper understanding of $AT_1$ receptor activation at an atomic level and suggest avenues for the design of allosteric $AT_1$ receptor modulators with a broad range of applications in GPCR biology, biophysics, and medicinal chemistry.

[1] College of Pharmacy, Ningxia Medical University, Yinchuan, Ningxia Hui Autonomous Region, China. [2] State Key Laboratory of Oncogenes and Related Genes, Key Laboratory of Cell Differentiation and Apoptosis of Chinese Ministry of Education, Shanghai Jiao Tong University, School of Medicine, Shanghai, China. [3] The CAS Key Laboratory of Receptor Research, Shanghai Institute of Material Medica, Chinese Academy of Sciences, Shanghai, China. [4] University of Chinese Academy of Sciences, Beijing, China. [5] Department of Biochemistry and Molecular Biology, Key Laboratory Experimental Teratology of Chinese Ministry of Education, School of Medicine, Shandong University, Jinan, Shandong, China. [6] Department of Hepatic Surgery VI, Eastern Hepatobiliary Surgery Hospital, Second Military Medical University, Shanghai, China. [7] Department of Cardiology, Renji Hospital, Shanghai Jiao Tong University, School of Medicine, Shanghai, China. [8] School of Pharmaceutical Sciences, Zhengzhou University, Zhengzhou, China. [9]These authors contributed equally: Shaoyong Lu, Xinheng He, Zhao Yang, Zongtao Chai, Shuhua Zhou. ✉email: lushaoyong@sjtu.edu.cn; sunjinpeng@sdu.edu.cn; jian.zhang@sjtu.edu.cn

As the largest superfamily of cell surface proteins in the human genome, G protein-coupled receptors (GPCRs) represent the therapeutic targets of nearly one-third of all approved drugs[1]. These receptors share a conserved structural architecture of seven transmembrane (7TM) helices linked by three extra- (ECLs) and three intracellular loops (ICLs)[2]. The GPCR-mediated signal transduction is always triggered by an extracellular signal to the orthosteric site located in the extracellular region of the 7TMs bundle center, which then transduces the stimuli to the intracellular region, thereby leading to the engagement of the receptor with G proteins or β-arrestins[3,4]. In addition, some GPCRs can also transmit signals in the absence of an external stimulus or an agonist, through 'basal' (also known as 'constitutive') activity[5–8]. The orthosteric site is conserved across the members of a single GPCR subfamily and, thus, poses a significant challenge in the development of selective drugs that can bind to a unique receptor subtype. As an alternative strategy, targeting a binding site outside the conserved orthosteric site, also termed as an "allosteric site", may provide avenues for the design of modulators with desirable selectivity profiles[9–14], which is a long-standing bottleneck in GPCR drug discovery.

Recent technological breakthroughs in structural biology, such as cryo-electron microscopy (cryo-EM) or X-ray free-electron lasers, have led to the identification of increasing GPCR structures[15–17], either in the inactive or active conformations, thereby providing mechanistic insights into the agonist-dependent receptor activation mechanisms that are useful for investigating structure-based drug design[18,19]. However, these high-resolution structures represent static snapshots obtained under specific experimental conditions; hence, they may miss important information pertinent to the conformational ensemble of GPCRs, as the receptors may have undergone a large-scale conformational transition during their (de)activation process[20,21]. Therefore, mechanistic and structural elucidation of the (de) activation pathway of the GPCRs is of paramount importance as distinct conformational states, such as intermediate, metastable, or transient states, present during the inactive-to-active transitions (or vice versa) of the receptors, are desired for a rational design of selective modulators.

The angiotensin II (AngII) type 1 receptor (AT$_1$ receptor), a prototypical class A GPCR, offers an important model for mechanistic exploration as it is a prominent therapeutic target for hypertension and related cardiovascular diseases[22]. Moreover, its high-resolution structures in both inactive, antagonist-bound[23] and active, agonist-bound, and nanobody-stabilized conformations have recently become available[24–26]. By comparing the structures of its inactive, antagonist-bound state (ZD7155; PDB ID: 4YAY)[23] (Fig. 1A) and the active state, meaning the AT$_1$ receptor complexed with a partial agonist S1I8 peptide, and a G protein mimetic nanobody to maintain the active conformation (PDB ID: 6DO1)[24] (Fig. 1B), the most remarkable differences in the two structures are found in the TMs 5–7 and helix 8 (H8) (Fig. 1C). Notably, on the intracellular side, the active state structure exhibits an outward displacement of TM5 and TM6, inward movement of TM7, and substantial repositioning of H8 parallel to the membrane, relative to the inactive state structure (Fig. 1C). On the extracellular side, the major conformational changes include an inward shift of TM5 and TM7 in the active state structure compared with the changes in the inactive state structure. Although the static, active, and inactive states of AT$_1$ receptor exhibit marked structural divergences, it has been challenging to completely capture the large-scale conformational transitions along the activation pathway of the AT$_1$ receptor experimentally. Therefore, it remains unclear how a dynamic pathway connects the inactive-to-active conformational transitions of the AT$_1$ receptor, thereby hindering a deeper

understanding of the comprehensive landscape of the activation mechanisms of this receptor as well as for other GPCRs. Furthermore, despite the availability of the inactive and active structures of the AT$_1$ receptor, there are no allosteric modulators of this receptor reported to date, suggesting a challenge for targeting potential allosteric binding sites in the two available snapshots. However, a cryptic allosteric site may exist in the transition pathway. Thus, it is advisable to capture key conformational substates along the activation pathway for the purpose of allosteric drug design[27,28].

To uncover the activation pathway of GPCRs, biosensors, nuclear magnetic resonance (NMR), and computational methods have been widely applied[29–36]. Among these approaches, molecular dynamics (MD) simulations have become a well-established technique for probing the conformational landscapes at an atomic level and directly uncovering biomolecular mechanisms[37–39]. Integrating MD simulations with Markov state models (MSMs)[40] has proven successful for understanding the molecular switches in β$_2$ adrenergic receptor (β$_2$AR)[41], elucidating ligand-driven conformational changes in CC chemokine receptor (CCR) 2[42], and revealing a cryptic pocket in dopamine D$_3$ receptor[43]. In the best structurally and biochemically characterized GPCRs, the rhodopsin receptor, the activation pathway, and the corresponding intermediate states have been elucidated by NMR[44], Fourier transform infrared spectroscopy[45], and MD simulations[46].

Here, we use a computational framework including a transition pathway generation algorithm, extensive all-atom MD simulations (300 μs) of the AT$_1$ receptor in the membrane-embedded environment, and MSM analysis for investigating the conformational landscapes of AT$_1$ receptor activation. We found an intermediate state during activation and identified a cryptic allosteric pocket on it. Multiple mutagenesis experiments confirm both the intermediate state and the potential regulation ability for the pocket. Our study not only offers a deep atomic-level insight into AT$_1$ receptor activation, but also provides an opportunity for the design of allosteric AT$_1$ receptor modulators.

## Results

**Extensive unbiased MD simulations reveal the activation pathway of AT$_1$ receptor.** To understand the inactive-to-active conformational transition pathway of the AT$_1$ receptor, we first generated a minimum energy path (MEP) by connecting the starting, antagonist-bound inactive (PDB ID: 4YAY), and the end, both nanobody- and agonist-bound fully active (PDB ID: 6DO1) states, by inserting a series of replicas between the two AT$_1$ receptor structures using the string method with the nudged elastic band (NEB) (see "Methods"). Both the nanobody and agonist were removed from 6DO1 and the antagonist was excluded from 4YAY. Here, we defined the GPCR structures with only agonist-bound forms as active conformations and with both agonist- and G protein-, β-arrestin-, or nanobody-bound forms as fully active conformations. After a simulated annealing process, 15 initial structures distributed on the MEP were selected and subsequently embedded with a 1-palmitoyl-2-oleoyl-sn-glycero-3-phosphocholine (POPC) membrane and explicit water. Each structure had 2 μs × 10 independent runs with random initial velocities, leading to a cumulative simulation timescale of 300 μs. Such an extensive timescale has been proven efficient for exploring the GPCR activation process[38]. As shown in Fig. 1, the most remarkable intracellular transmembrane domain conformational variations of the AT$_1$ receptor during activation are arguably the outward movements of TM5 and TM6, and the inward displacement of TM7. Thus, we defined two activation parameters (Fig. 1C) to project the simulated trajectories onto a

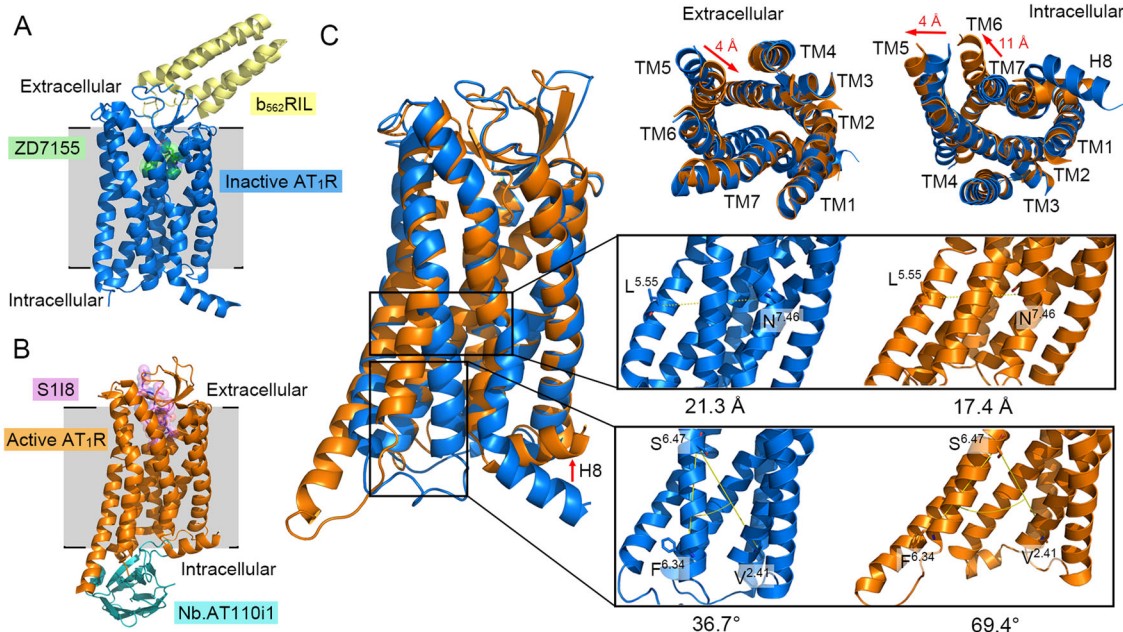

**Fig. 1 The distinction between the inactive and fully active AT$_1$ receptor crystal structures. A** The overall structure of the inactive AT$_1$ receptor in complex with an antagonist ZD7155 (green) and b$_{562}$RIL (yellow). **B** The overall structure of the fully active AT$_1$ receptor in complex with a partial agonist S1I8 (magenta) and a G protein mimetic nanobody (cyan). **C** Major conformational changes between the inactive (blue) and fully active (orange) AT$_1$ receptor. Red arrows show obvious transmembrane (TM) movements during activation. The zoom-in views represent the distance between the Cα atoms of L$^{5.55}$ and N$^{7.46}$ and the angle among the Cα atoms of F$^{6.34}$, S$^{6.47}$, and V$^{2.41}$, in order to monitor the conformational rearrangements of TMs 5–7. The superscripts refer to the Ballesteros–Weinstein numbering system. All structure figures were drawn by PyMOL. AT$_1$R angiotensin II type 1 receptor, S1I8 S1I8 mutant angiotensin II, Nb Nanobody, TM transmembrane helices, H8 helix 8.

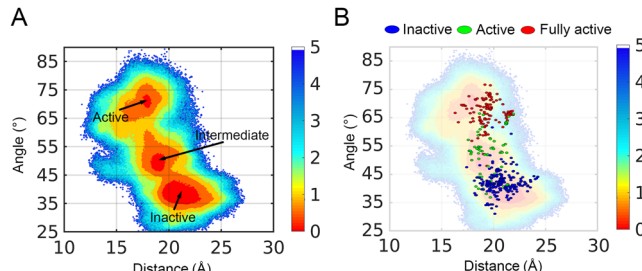

**Fig. 2 The free-energy landscape of AT$_1$ receptor. A** Conformational landscape of the AT$_1$ receptor generated using the Cα atom distance between L$^{5.55}$ and N$^{7.46}$, and the angle among the Cα atoms of F$^{6.34}$, S$^{6.47}$, and V$^{2.41}$ as the order parameters along the activation pathway. Arrows point out different states on the landscape. Color scales of the landscape are shown on the right. MATLAB is applied to draw the landscape. **B** Projection of all reported fully active (red), inactive (blue), and active (green) structures of class A GPCRs onto the AT$_1$ receptor conformational landscape. The unit of free-energy values is kcal/mol.

two-dimensional (2D) space to comprehensively capture the conformational landscape of AT$_1$ receptor activation. Notably, L$^{5.55}$ (superscripts indicate the Ballesteros–Weinstein numbering for GPCR residues)[47] undergoes a large and conserved rearrangement during GPCR activation and N$^{7.46}$ localizes at the TM7 twist position, which reflects the inward movement of TM7 during activation. Thus, one parameter for defining activation was the distance between the Cα atoms of L$^{5.55}$ and N$^{7.46}$, which represented the conformational changes of TM5 and TM7. The other parameter was the angle among the Cα atoms of F$^{6.34}$, S$^{6.47}$, and V$^{2.41}$, which reflected the outward movement of TM6, a crucial hallmark of class A GPCR activation that provides space to accommodate downstream signal proteins including G

proteins or β-arrestins[48]. As shown in Fig. 1C, the corresponding distance and angle values were markedly distinct between the inactive and fully active AT$_1$ receptor crystal structures, thereby highlighting the discriminatory power of the activation parameters.

Based on these activation parameters, we calculated the corresponding value of each snapshot during the simulations and plotted a free-energy landscape (Fig. 2A), depicting the inactive-to-active transition pathway of AT$_1$ receptor activation. Since the initial inactive crystal structure of the AT$_1$ receptor was located at a distance and angle of 21.3 Å and 36.7°, respectively, the largest free-energy basin with a distance of ~19–23 Å and an angle of ~35–42° represented the inactive region. Further, owing to the outward movement of TM6, the angle gradually increased, reflecting an open degree of TM6, whereas the movement of TM5 and TM7 caused a decrease in the interhelical distance. During the activation process, the AT$_1$ receptor overcomes a relatively low-energy barrier and enters the intermediate state located at a free-energy basin with a distance of ~17–19 Å and an angle of ~46–51°, and it then crosses a high energy barrier at ~17 Å and 63° to arrive at the fully active state (17.4 Å, 69.4°), representing the coordinates of the fully active AT$_1$ receptor crystal structure (PDB 6DO1). Overall, the convergence of the landscape was proven in both the timescale of a single trajectory and the number of rounds (Supplementary Note 1), confirming that the sampling has been sufficient to explore AT$_1$ receptor activation.

Because of the recent breakthroughs in receptor crystallization, more than 440 class A GPCR structures of >70 receptors have been determined by X-ray or cryo-EM crystallography (Supplementary Note 2 and Supplementary Data 1). These structures represent the GPCR conformational ensemble covering all inactive, active, and intermediate states, including different inverse agonist- or antagonist-bound inactive conformations, agonist-bound active conformations, and their effector G

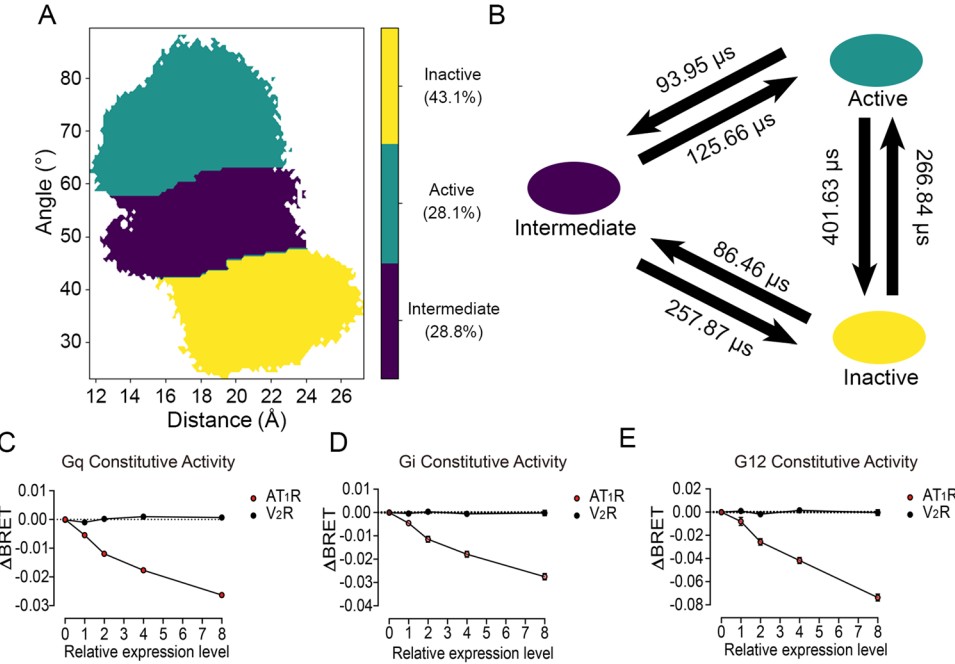

**Fig. 3 The three macrostates divided by MSM, and constitutive activity. A** The distribution of active (green), inactive (yellow), and intermediate (purple) states on the free-energy landscape. The attribution and probability of each macrostate are shown on the right. **B** The transition time among the active (green), inactive (yellow), and intermediate (purple) states, represented by the mean first passage time. Constitutive activities of AT$_1$ receptor in Gq (**C**), Gi (**D**), and G12 (**E**) pathways (red). Vasopressin 2 receptor (V$_2$R) was used as a negative control (black). The gradient cell surface expression levels of AT$_1$ receptor and V$_2$R were achieved by adjusting the transfecting amounts of plasmids encoding the respective receptor in HEK293 cells (Supplementary Fig. 3). Data were from three independent experiments. The bars indicate the mean ± SEM values. The absolute luminescence intensity values are provided in source data. If no special instructions were shown, histograms were drawn by GraphPad. ΔBRET: the change of bioluminescence resonance energy transfer value.

protein-, β-arrestin-, or nanobody-stabilized fully active conformations. Considering the conformational heterogeneity of class A GPCR structures, we projected these experimentally solved structures onto the 2D conformational landscape of the AT$_1$ receptor sampled by the simulations. The corresponding residues at positions 5.55, 7.46, 6.34, 6.47, and 2.41 of each structure were selected to calculate the distance and angle using their Cα atoms. Then, the fully active, inactive, and active states were mapped to provide insights into the dynamic conformational landscape of the receptors. The projection suggests that the AT$_1$ receptor activation pathway samples a wide conformational landscape of GPCR ensemble with the major distribution of experimental structures within the reaction path (Fig. 2B). This result suggests that the computational model is in reasonable agreement with the experimental data, highlighting the ability of our simulations to reproduce the overall class A GPCR activation and suggesting that it is thus suitable for further investigations.

Interestingly, the inactive energy basin largely matched its crystal structures. This can perhaps be ascribed to the fact that the inactive structures have mostly similar TM5−TM7 conformation. The outliers—platelet-activating factor receptor (PDB 5ZKP) and P2Y$_{12}$ receptor (PDB 4NTJ)—however, have an unusual anchor (TM2) movement and distinct TM6 twist, respectively (Supplementary Note 3)[49,50]. Since agonist binding shifts the receptor conformational ensemble toward the active state, resulting in increased conformational dynamics and the coexistence of multiple states along the activation pathway[51,52], diverse structurally activated states can be captured under distinct conditions. For example, with the same ligand AM-841, the type 1 and 2 cannabinoid receptors (CB$_1$R and CB$_2$R; PDB 5XR8 and 6KPC, respectively) sample entirely different conformations. After binding another agonist, CP55940, and a negative allosteric

modulator, ORG27569, CB$_1$R (PDB ID: 6KQI) also situates at the inactive cloud (Supplementary Note 3). Indeed, half of the active structures are located at the intermediate cloud sampled by the simulation, indicating that the intermediate state extensively exists in class A GPCRs. Most fully active structures are proximal to the active cloud sampled by the simulation, except the rhodopsin receptor that has a larger distance (~22 Å) between TM5 and TM7 due to its unique activation mechanism[45] (Supplementary Note 3). Of note, receptor-Gs complex has a larger angle than receptor-Gi complex, which is consistent with the different volumes of the two downstream proteins. The position of the G protein complex has a similar distribution with nanobody-stabilized structures, suggesting that the nanobodies used in crystallization have a minor influence on its conformations (Supplementary Note 4). Since our simulations are based on the apo structure, the active cloud is not highly correlated with fully active non-rhodopsin structures, indicative of the instability of active structures and the necessity for transducers to stabilize the fully active conformations. Collectively, the free-energy landscape illustrates the activation process of the AT$_1$ receptor and uncovers a hidden intermediate state along the pathway linking the inactive and active receptors.

**Markov state model discovers a synergistic transition mechanism of AT$_1$ receptor activation.** To elucidate the mechanism of AT$_1$ receptor activation, we built a kinetic network MSM using the activation parameters. From a statistical viewpoint, MSM provides summarized insights of the conformational ensemble of biomacromolecules at equilibrium. Therefore, the key intermediate states are identified more precisely and the thermodynamic properties, such as transition timescale, are also quantified[40,41]. Upon validation of the Markovian properties by

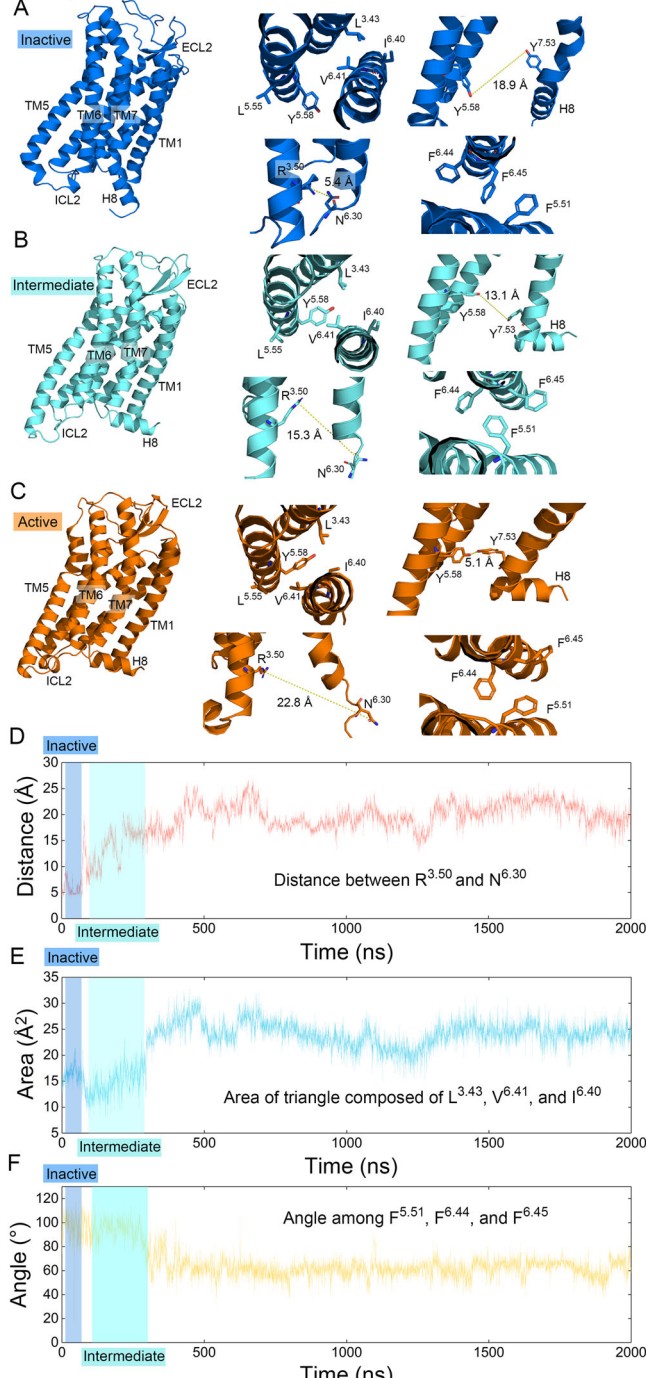

**Fig. 4 The synergistic activation mechanism of AT$_1$ receptor. A–C** The comparison of representative inactive (A, blue), intermediate (B, cyan), and active (C, orange) structures. Zoom-in views of key structures are depicted on the right of (**A**), (**B**), and (**C**). Top left: the breaking of hydrophobic lock during activation. Top right: the activation renders the approach of Y$^{7.53}$ and Y$^{5.58}$. Bottom left: polar interaction on the position of common ionic lock is broken on the process of activation. Bottom right: F$^{6.44}$ and F$^{6.45}$ ratchet past F$^{5.51}$ with the outward movement of TM6. Distances between atoms are shown in dashed yellow lines. Dynamic variations of these activation properties during a representative simulation trajectory are shown in (**D**) for the distance between the guanidine carbon atom of R$^{3.50}$ and the Cγ atom of N$^{6.30}$ (red), (**E**) for the area of the triangle composed of the Cγ atom of L$^{3.43}$, the Cβ atom of V$^{6.41}$, and the Cβ atom of I$^{6.40}$ (blue), and (**F**) for the angle among the centroids of the benzene ring of F$^{5.51}$, F$^{6.44}$, and F$^{6.45}$ (yellow).

implied timescale test (Supplementary Note 5), we constructed a 200-microstate MSM using the k-means algorithm and clustered them into three macrostates (Fig. 3A), using the Robust Perron Cluster Cluster Analysis (PCCA+) algorithm. Next, transition path theory (TPT) was applied to map out the transitions and their timescales among the active, inactive, and intermediate states (Fig. 3B). The prediction of MSMs in the three macrostates was also confirmed by the Chapman–Kolmogorov test (Supplementary Note 5).

The conformational space was divided into three regions, and the locations of yellow, green, and purple regions corresponded to inactive, active, and intermediate states, respectively (Fig. 3). Thus, the population ensemble determined by the 3-macrostate MSM was consistent with our state distribution. Notably, the proportion of the inactive state (43.1%) was the largest in the ensemble, in agreement with the fact that the apo AT$_1$ receptor naturally stays in the inactive state. However, the active state also occupied 28.1% of the conformations during simulation, indicating that the apo AT$_1$ receptor still has the ability to trigger downstream pathways and the AT$_1$ receptor might have constitutive activity.

The transition from the inactive to the active states (401.63 μs) was significantly longer than the transition from the active to inactive states (266.84 μs), as shown in the transition timescale (Fig. 3B). This observation supports the concept that the apo AT$_1$ receptor is normally inactive, and that the active conformer also exists in its ensemble. Based on the AT$_1$ receptor conformational distribution, it is evident that both the active (86.46 μs) and inactive (93.95 μs) states are amenable to transfer to the intermediate state. However, it was difficult for the intermediate state to transfer to the active (257.87 μs) and inactive (125.66 μs) states. Hence, the kinetics of the activation pathway indicates that the rate-limiting step of AT$_1$ receptor activation is the transition from the intermediate to the active states.

As our simulations were based on the apo AT$_1$ receptor, the spontaneous activation reflects a constitutive activity of the wild-type (WT) AT$_1$ receptor, which has been previously reported[53,54]. Using bioluminescence resonance energy transfer (BRET) assays in G proteins, we confirmed the constitutive activity of the AT$_1$ receptor in Gq, Gi, and G12 compared with a control vasopressin 2 receptor (V$_2$R). We first controlled the equal levels of the AT$_1$ receptor and V$_2$R on the cell surface (Supplementary Note 6). Next, the dissociation of Gα and Gβγ was represented by the decrease of ΔBRET. The ΔBRET decreased with increasing AT$_1$ receptor levels in a dose-dependent manner, suggesting that downstream G proteins are dissociated from the apo AT$_1$ receptor (Fig. 3C–E). In contrast, V$_2$R levels had no influence on the ΔBRET value. Collectively, these data suggest a constitutive activity of the AT$_1$ receptor.

To further elucidate the activation mechanism of the AT$_1$ receptor, we evaluated key residue rearrangements. During GPCR activation, the emergence of residue rearrangements—or micro-switches—is fundamental to TM movement[48]. Common activation micro-switches include breaking of the hydrophobic lock composed of residue positions 3.43, 6.40, and 6.41, which loosens the connection between TM3 and TM6, and the rupture of a conserved ionic lock between residue positions 3.50 and 6.30, which releases TM6 outward[48,55]. In addition, the AT$_1$ receptor harbors unique residue movements as well. In the inactive state, R$^{3.50}$ and N$^{6.30}$ form a relatively weak polar interaction instead of the conserved ionic lock and break upon activation. Further, F$^{6.44}$ and F$^{6.45}$ ratchet over F$^{5.51}$ when the AT$_1$ receptor is activated. In addition, Y$^{7.53}$ and Y$^{5.58}$ tend to approach each other during AT$_1$ receptor activation, which reflects the movement of TM5 and TM7 (ref. [24]).

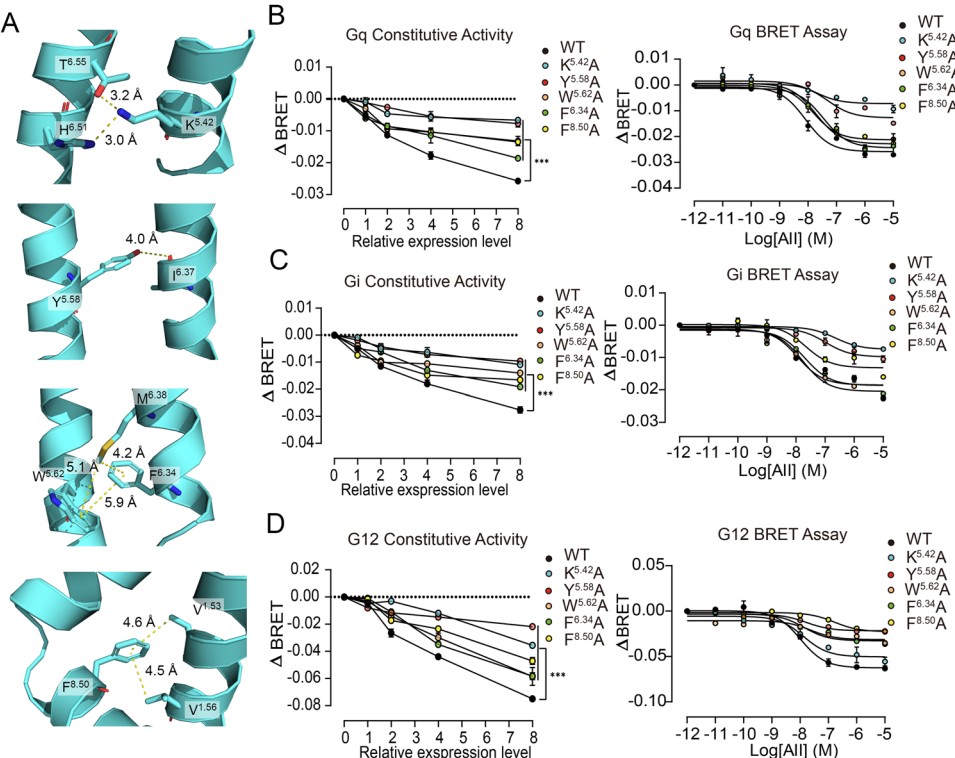

**Fig. 5 The intermediate-specific micro-switches and their importance for activation. A** The intermediate-specific micro-switches. Involved residues are shown in sticks and corresponding distances are depicted by yellow dashed lines. **B–D** Constitutive and AngII-induced activities of WT AT$_1$ receptor and mutants in Gq (**B**), Gi (**C**), and G12 (**D**) pathways. WT, K$^{5.42}$A, Y$^{5.58}$A, W$^{5.62}$A, F$^{6.34}$A, and F$^{8.50}$A mutations are shown as black, cyan, red, orange, green, and yellow points. Data were from three independent experiments and representative dose-response curves were shown in AngII-induced activity. The summary of data was shown in Supplementary Note 14. The absolute luminescence intensity values are provided in source data. ΔBRET: the change of bioluminescence resonance energy transfer value. ***$P < 0.001$; HEK293 cells transfect with AT1R mutants compared with those transfected with WT AT1R. The bars indicate the mean ± SEM values. Statistical differences between WT and mutants were analyzed using two-way ANOVA with Dunnett's post hoc test. (The $p$ values in **B–D** are <0.0001, <0.0001, <0.0001).

We then clustered the representative conformations of the macrostates according to the root mean square deviation (RMSD) between structures (see "Methods"). The micro-switches are also shown in the inactive, intermediate, and active AT$_1$ receptor states (Fig. 4A–C). Based on a representative simulation trajectory across the three states, we observed the dynamic positioning of our switches (Fig. 4D–F). Globally, TM6 moved outward, while TM5 and TM7 approached the inactive-to-active conformational transition. Meanwhile, the landscape coordinates of the inactive, intermediate, and active representative conformations were located at (23.3 Å, 40.3°), (19.0 Å, 50.6°), and (17.1 Å, 71.3°), respectively; matching the low-energy conformer and further strengthening the accuracy of our cluster methodology.

Other structural elements also appear functionally significant. Upon activation, ECL2 and extracellular TM6 moved toward the center of the endogenous AngII pocket and nearly closed the pocket in the active state, indicating that ECL2 and TM6 stabilize the pocket and maintain the binding of AngII during activation (Supplementary Note 7). The closure of the orthosteric pocket has also been observed in the active β$_2$AR-nanobody complex structure[56]. Moreover, intracellular TM5 formed more α-helices in the active state, and the length of ICL2 decreased accordingly to help the binding of the downstream proteins, which is in line with the common GPCR activation process. In addition, H8 moved upward from the inactive to the active state to provide space for the downstream effectors. This upward movement produces a suitable accommodation for ligand binding among TM1, TM7, and H8, which may offer an opportunity for the formation of a cryptic pocket for drug design (Supplementary

Note 7). We further explored the variation of secondary structural elements among macrostate trajectories using the Definition of Secondary Structure of Proteins algorithm (Supplementary Note 8) and community analysis for the signal transfer (Supplementary Note 9). During receptor activation, the ECL2 became stable, the intracellular TM5 was elongated, and connections to the transducer pocket increased. The upward movement of H8 was also observed.

In agreement with the observations revealed by comparing the crystal structures, the micro-switches generally and synergistically changed from the inactive state to the intermediate and active states. For example, V$^{6.41}$ moved toward the membrane step-by-step and finally broke the hydrophobic lock in the active structure. This movement provided space for Y$^{7.53}$, which finally approached Y$^{5.58}$ (inactive: 18.9 Å, intermediate: 13.1 Å, active: 5.1 Å). In addition, the distance between R$^{3.50}$ and N$^{6.30}$ increased upon the outward movement of TM6 (inactive: 5.4 Å, intermediate: 15.3 Å, active: 22.8 Å), which in turn, rendered F$^{6.44}$ and F$^{6.45}$ to ratchet over F$^{5.51}$. Meanwhile, the relative positions of the residues in the intermediate state were in between the active and inactive states, indicating that the activation process generally occurs. In a single spontaneous activation trajectory (Fig. 4D–F), the variations of the three micro-switches—breaking of the ionic lock (Fig. 4D), the opening of the hydrophobic lock (Fig. 4E), and rearrangement of phenylalanine ratchet (Fig. 4F)—synchronize, and thus reflect the synergistic activation mechanism of the AT$_1$ receptor. The rationale for the measurement of residue pairs describing the hydrophobic lock is provided in Supplementary Note 10. Thus, the zoom-in views of the residue

rearrangements were consistent with the global secondary structural movements.

To demonstrate the transition pathway, we employed gaussian accelerated MD (GaMD) simulations on the holo AT₁ receptor systems using the intermediate structure as a model system. The inverse agonist olmesartan and the endogenous agonist AngII were docked to the orthosteric pocket of the intermediate state, respectively. Then, the two holo systems underwent 2 μs conventional MD simulations followed by 1 μs GaMD simulations for three independent runs. In the corresponding free-energy landscapes, AngII binding pushed the open of TM6 along with the inward movement of TM5 and TM7, resulting in the active state, while olmesartan binding led to a smaller TM6 angle and a larger TM5–TM7 distance, shifting the receptor conformation to the inactive state (Supplementary Note 11). Together, these results suggest the likely existence of an intermediate structure as captured from the transition pathway.

Since the AT₁ receptor activates multiple G protein subtypes in its apo and holo states, we further explored whether this intermediate state could be commonly required by different G protein downstream signal pathways. By comparing the intermediate structure with the inactive and active macrostate structures, we identified several specific micro-switches for the intermediate state, such as polar contacts among $K^{5.42}$, $H^{6.51}$, and $T^{6.55}$; a hydrogen bond between $Y^{5.58}$ and $I^{6.37}$; hydrophobic network among $M^{6.38}$, $W^{5.62}$, and $F^{6.34}$; and hydrophobic contacts among $V^{1.53}$, $V^{1.56}$, and $F^{8.50}$ (Fig. 5A and Supplementary Note 12). Based on these intermediate-specific interactions, we designed $K^{5.42}A$, $Y^{5.58}A$, $W^{5.62}A$, $F^{6.34}A$, and $F^{8.50}A$ mutations to specifically disrupt the intermediate state. The protein levels of the variants were similar to that of the WT receptor (Supplementary Note 13). Constitutive and AngII-induced activities for G proteins were then measured using BRET assays to evaluate the influence of these mutations (Fig. 5B–D).

Based on BRET assays, the disruption of micro-switch interactions in the intermediate state inhibited both the constitutive and AngII-induced activities of the AT₁ receptor for G proteins, including Gq, Gi, and G12. The decrease of constitutive activity was more obvious than the AngII-induced activity. In the AngII-induced activity, $K^{5.42}A$ decreased the activity of Gq and Gi more than G12, suggesting that Gq and Gi activity may need a larger movement of extracellular TM5. Although $W^{5.62}A$ and $F^{6.34}A$ did not influence the Gq and Gi signal in $E_{max}$, the increase of the $EC_{50}$ value reflected a weaker activation upon AngII binding (Supplementary Note 14). The remaining $Y^{5.58}A$ and $F^{8.50}A$ mutations similarly modulated the activity of the AT₁ receptor in the three G proteins. In summary, these results indicated that the intermediate state is required for all G protein signaling pathways, highlighting its possibility to become a drug target.

**Distinct active conformations from global movement can induce biased signals.** Apart from the dimension reduction accomplished by the features extracted from the biological process (such as the activation parameters shown in Fig. 1C), we also applied time-structure-based independent component analysis (tICA) to our system to analyze the global movement of the receptor during activation. tICA employs linear combinations to particular features, such as the phi/psi angle of the backbone, to maximize the decorrelation time among these features. Thus, tICA enables the capture of slow dynamic processes during simulations[40,57]. With the help of tICA, we projected the trajectories onto another 2D landscape, which was representative of the global phi/psi movement. Through the implied timescale test, dimensionality reduction using tICA has been proven to have

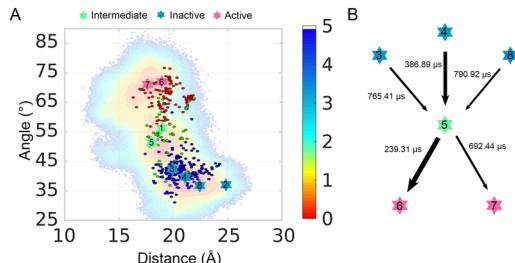

**Fig. 6 The eight macrostates clustered by tICA. A** The projection of eight macrostates on the landscape based on their Cα distance between $L^{5.55}$ and $N^{7.46}$, and the angle among the Cα atoms of $F^{6.34}$, $S^{6.47}$, and $V^{2.41}$. Color scales of the landscape are shown on the right. **B** The transition time between the major six structures whose proportions are the most, presented in the form of the activation pathway. The thickness of arrows reflects the transition time. Red, blue, and green stars represent active, inactive, and intermediate macrostates in the two subplots.

Markovian properties. The tICA landscape was separated into eight macrostates according to the PCCA+ algorithm, and its prediction met the requirements of the Chapman–Kolmogorov test (Supplementary Note 15). To analyze the global movement during simulations, the representative conformation of each tICA macrostate was extracted and projected onto the distance-angle landscape based on their corresponding activation parameters (Fig. 6A). Finally, the TPT method was applied to calculate the transition dynamics between six major macrostates (Fig. 6B).

As shown in Fig. 6A, macrostates 1 and 5 were in the intermediate state, macrostates 2–4 and 8 were in the inactive state, and macrostates 6 and 7 were in the active state. Notably, the macrostates 3–8 occupied more than 96% of the snapshots during our simulations (Supplementary Note 15) and stayed close to the energy basins in our landscape (Fig. 2A), thereby suggesting that the coordinate parameters (distance and angle) elucidated the activation process just as the global movement did. Thus, the mechanism according to the activation parameters is credible. In addition, the representative conformations of macrostates 4 and 6 corresponded to the area with the most inactive and active crystal structures, respectively (Figs. 2B and 6A). Pathway 4–5–6 also takes the shortest transition time, thus it is the most preferential way to activate the AT₁ receptor among all the analyzed pathways (Fig. 6B). Hence, macrostates 4 and 6 are crucial conformers along the activation pathway, that is, macrostate 4 is encountered before entering the activation process, and the typical activation process involves the macrostates 4–6.

Since the AT₁ receptor has a constitutive activity for G proteins (Fig. 3C–E) and shows β-arrestin activity upon AngII binding or specific agonists[25], we explored the connection between different active states and biased signaling. Because Gq is the major G protein activated by the AT₁ receptor and β-arrestin 2 is commonly used in biased signaling[38,58,59], we first constructed AT₁ receptor−Gq and AT₁ receptor−β-arrestin 2 complexes based on the active macrostates 6 and 7 to determine whether the two active conformations had a bias for these transducers (Supplementary Note 16). The models showed that macrostates 6 and 7 tend to initiate Gq and β-arrestin 2, respectively, suggesting that biased conformations naturally exist in the ensemble of the AT₁ receptor.

Supported by this preliminary analysis, we further investigated biased signaling in the AT₁ receptor based on tICA analysis. From the representative structures, we explored specific micro-switches of each macrostate (Fig. 7A). $Y^{7.53}$ is close to hydrophobic residues at TM1 and TM2 in both macrostates 5 and 6, while it

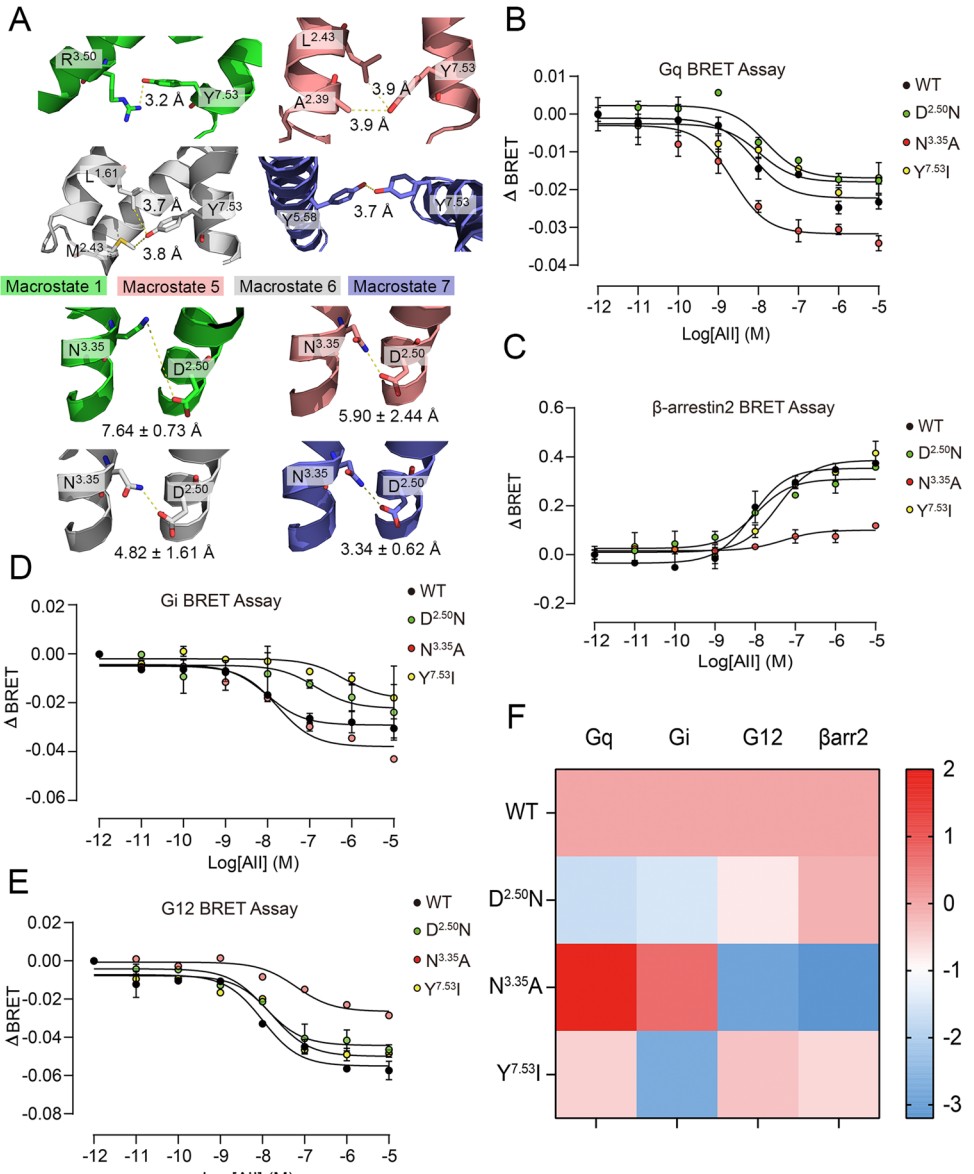

**Fig. 7 Different tICA macrostates are responsible for different biased signaling. A** Macrostates 1 (green), 5 (salmon), 6 (gray), and 7 (purple) with their corresponding specific micro-switches are shown in cartoon. Involved residues are shown in sticks. Because of the obvious fluctuation of the distance between N[3.35] and D[2.50] in representative trajectories of macrostates, the distance is shown in mean ± standard deviation form for all snapshots from the trajectories. AngII-induced Gq activation (**B**), β-arrestin 2 recruitment (**C**), Gi activation (**D**), and G12 activation (**E**) in HEK293 cells transfected with WT AT$_1$ receptor (black) or D[2.50]N (green), N[3.35]A (red), and Y[7.53]I (yellow) mutants. Three independent experiments were performed and representative dose-response curves were shown. The bars indicate the mean ± SEM values. **F** Heat map of AT$_1$ receptor signaling signatures of WT AT$_1$ receptor and mutants. Blue and red squares represent decreased or increased activity compared with WT AT$_1$ receptor. The relative activities Δlog($\tau/K_A$) of WT AT$_1$ receptor and mutants in each signaling pathway calculated in (Supplementary Note 17) were expressed as a heat map. Color scales of the map are shown on the right. ΔBRET: the change of bioluminescence resonance energy transfer value.

forms a hydrogen bond with R[3.50] or Y[5.58] in macrostates 1 and 7, respectively. Thus, by introducing a hydrophobic residue, the Y[7.53]I mutation could maintain the hydrophobic interactions, but disrupted the polar interactions, leading to the stability of macrostates 5 and 6, and the instability of macrostates 1 and 7. In addition, N[3.35] and D[2.50] form a tight hydrogen bond in macrostate 7, whereas it is weak in macrostate 6 and diminishes in macrostates 1 and 5. N[3.35]A mutation disrupted the hydrogen bonding interaction, preferred the conformation with a long distance between the residue at 3.35 and the polar residue at 2.50 (macrostates 1, 5, and 6). In turn, a moderate mutation D[2.50]N changes the charge of its sidechain, which may weaken the

interaction between residue at 3.35 and residue at 2.50. Because macrostate 7 has a strong hydrogen bond, its conformation may maintain in response to the D[2.50]N mutation. However, macrostate 1, 5, and 6 may be disturbed owing to this mutation. Our mutation experiments for biased signaling were benchmarked against these micro-switches and the corresponding results (Fig. 7B–E). An operational model was also applied to determine the biased signaling with normalized $E_{max}$ values (Fig. 7F).

In Fig. 7F, blue and light pink represent a weak signal and red reflects a strong signal, compared with the WT AT$_1$ receptor. In particular, N[3.35]A mutation (benefits macrostates 1, 5, and 6,

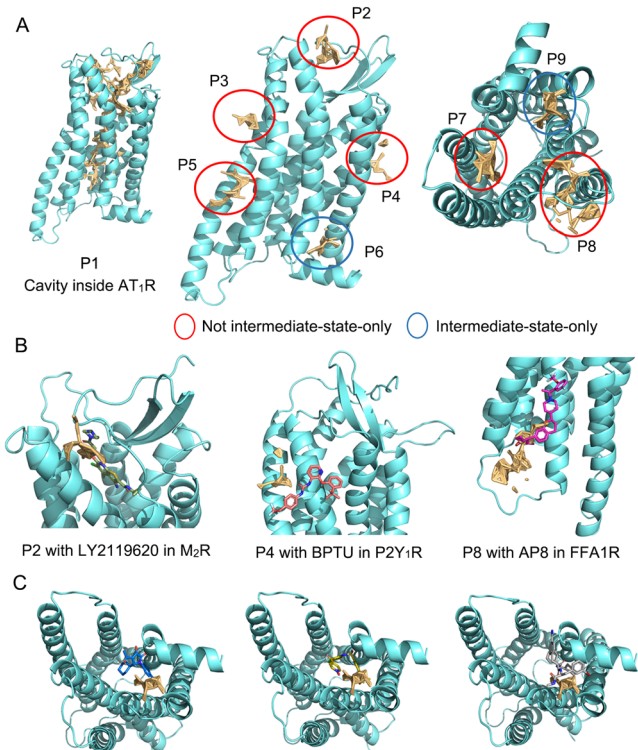

Red circle legend: ◯ Not intermediate-state-only ◯ Intermediate-state-only

P2 with LY2119620 in M₂R    P4 with BPTU in P2Y₁R    P8 with AP8 in FFA1R

P9 with CCR2-RA-[R] in CCR2    P9 with vercirnon in CCR9    P9 with Cmp-15PA in β₂AR

**Fig. 8 The predicted pockets on the intermediate state and comparison of them with other GPCR allosteric sites. A** All the pockets predicted by Fpocket in the intermediate state. Red circles show the overlap pockets with the prediction of inactive and active representative structures, while blue circles mean the intermediate-only pockets P6 and P9. **B** The alignment of current allosteric modulators on their sites with intermediate structure and pockets P2, P4, and P8. **C** The overlap of P9 with current allosteric GPCR modulators in crystal structures. In (**B**) and (**C**), cyan cartoons depict the intermediate AT₁ receptor and orange sticks show corresponding pockets defined by Fpocket. Sticks in other colors show the allosteric modulators. AT₁R: angiotensin II type 1 receptor.

inhibits macrostate 7) obviously promoted Gq and Gi signals but inhibited G12 and β-arrestin 2 signals. In contrast, $D^{2.50}N$ showed relatively weaker Gq and Gi signals compared with G12 and β-arrestin 2 signals (Fig. 7F and Supplementary Note 17). Facilitating macrostates 5 and 6, $Y^{7.53}I$ also promoted Gq but inhibited β-arrestin 2. Thus, micro-switches facilitating macrostate 6 and restraining macrostate 7 ($N^{3.35}A$, $Y^{7.53}I$) led to the Gq activation and β-arrestin 2 inhibition, whereas the $D^{2.50}N$ mutation, boosting macrostate 7 and reducing macrostate 6, preferred β-arrestin 2 rather than Gq signal. This is in line with our hypothesis that macrostate 6 is biased to Gq pathway, whereas macrostate 7 is for β-arrestin 2 (Supplementary Note 16). As for Gi, it can be inferred that the intermediate macrostate 1 (inhibited by $Y^{7.53}I$ and $D^{2.50}N$, facilitated by $N^{3.35}A$) may play a critical role in the activation toward Gi since $Y^{7.53}I$ and $D^{2.50}N$ suppressed Gi signal but $N^{3.35}A$ increased it. Conversely, G12 activation may be related to the macrostate 5 (facilitated by $Y^{7.53}I$, inhibited by $N^{3.35}A$ and $D^{2.50}N$) because $Y^{7.53}I$ stimulated the G12 signal, while $N^{3.35}A$ and $D^{2.50}N$ repressed G12 activation. The effect of mutations can be confirmed by recent structural information. For instance, $Y^{7.53}I$ quenches polar interactions with $R^{3.50}$ and $Y^{5.58}$, which releases intracellular TM3 and TM5. The flexible TM3 can move downward and increase the interaction of ICL2 with transducers, while TM5 might extend toward G protein. Accordingly, the increased ICL2

interaction and extended TM5 are properties of GPCR-Gq/11 complex compared with GPCR-Gi/o complex[60–62]. Thus, $Y^{7.53}I$ promotes Gq but inhibits Gi signals in the AT₁ receptor.

In summary, we observed hints for each biased signal pathway in our MD simulations for the apo AT₁ receptor and confirmed them by site-directed mutation experiments. During biased signaling, different ligands stabilize distinct receptor conformers, reflecting the biased signaling of the AT₁ receptor via a conformational selection mechanism. Taken together, tICA identified a transition pathway based on global movements, and the active and intermediate conformations provided indications of biased signaling.

**Identification of a cryptic allosteric site in the intermediate state.** We also used a pocket prediction algorithm to identify potential sites on macrostates and guide the allosteric drug design. Using Fpocket, we identified and clustered nine pockets on the hidden intermediate state (Fig. 3A) and several pockets in other states. Fpocket defines alpha spheres by Voronoi tessellation and regards the alpha spheres that are concentrated on clefts or cavities and with the ability to bind small molecules as the pocket[63]. In Fig. 8A and Supplementary Note 18, the positions of the pockets are depicted and the overlapping pockets between the intermediate state and other states are also highlighted. Most of the detected pockets in the intermediate state corresponded to pockets in the inactive and active states, except for P6 and P9, which were the cryptic pockets only in the intermediate state.

Among the nine pockets, P1 was located at the center of the AT₁ receptor and spanned from top to bottom. It revealed a large cavity inside the AT₁ receptor that partly overlapped with the orthosteric site for AngII. Nevertheless, P1 was too large for the design of small-molecule modulators. In addition, the P2–P6 pockets laid on the membrane side, whereas the intracellular P7–P9 pockets were in the proximity of the transducer site (Fig. 8A). The volume of P2–P9 pockets could fit possible allosteric modulator sites, and P2, P4, P8, and P9 matched previous allosteric drug sites as well (Fig. 8B, C). In particular, P2 was mostly identical to the LY2119620 pocket in the M₂ muscarinic acetylcholine receptor (PDB 4MQT)[64]. P8 also partially overlapped with the allosteric agonist AP8 pocket in free fatty acid receptor 1 (PDB 5TZY)[65]. In addition, BPTU interacted with P2Y₁ receptor (PDB 4XNV)[66] in the vicinity of P4, confirming the accuracy of our site prediction. P9 overlapped with three allosteric sites observed in GPCR-modulator structures as an intermediate-only potential pocket. The sites were as follows: the CCR2-RA-[R] site in CCR2 (PDB 5T1A), the vercirnon site in CCR9 (PDB 5LWE), and the Cmp-15PA site in β₂AR (PDB 5X7D), and were not confined to the intermediate state[67–69]. Thus, P9 is a common allosteric site in the intermediate state of the AT₁ receptor. However, since P2, P4, P8, and P9 emerged as the allosteric sites in other class A GPCR-modulator structures, the sites shown in Fig. 8A are highly likely allosteric sites.

Since P9 was excluded because of its universality, P6 remained the only cryptic allosteric site in the intermediate state. As P6 has never been reported in previous GPCRs and allosteric modulator structures, our findings indicate that it may be an allosteric site for drug design. We also conducted molecular screening on P6 using a GPCR allosteric compound library and found a binding pose highly overlapping with P6 from the best-scored ligand (Supplementary Note 19), further suggesting the targetable potential of P6.

**Mutagenesis and FlAsH-BRET assay confirm the predicted cryptic binding site.** Next, we confirmed that P6 was an allosteric

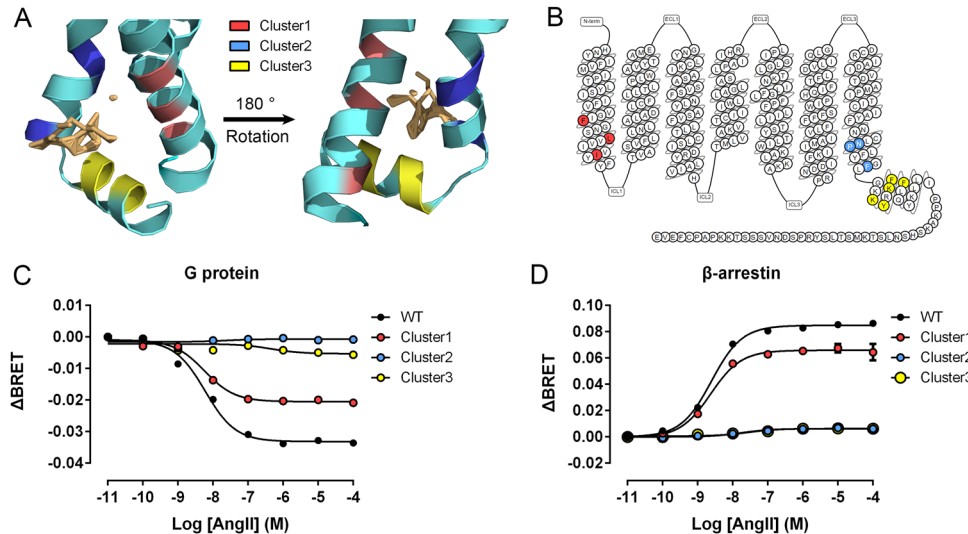

**Fig. 9 Cluster mutations confirm P6 as an allosteric site. A** The positions of clusters 1 (red), 2 (blue), and 3 (yellow) on the intermediate structure (cyan). **B** The sites of alanine-scanning clusters on the 7TMs of the $AT_1$ receptor. AngII-induced Gq activation (**C**) and β-arrestin 2 recruitment (**D**) through WT $AT_1$ receptor (black) and cluster 1 (red), 2 (blue), and 3 (yellow) mutants. Three independent experiments were performed, and representative dose–response curves were shown. The bars indicate the mean ± SEM values.

site firstly using clustered alanine-scanning mutagenesis. Three groups of mutations (Fig. 9A, B) were independently introduced to directly investigate the effect of allosteric perturbations of P6 on the transducer activity of the $AT_1$ receptor. The cluster 1 mutations included $F^{1.48}A$, $L^{1.52}A$, and $I^{1.57}A$, located on the intracellular side of TM1. The cluster 2 mutations were $N^{7.49}A$, $P^{7.50}A$, and $F^{7.55}A$ on TM7. The cluster 3 mutations ($K^{8.49}A$, $F^{8.50}A$, $K^{8.51}A$, $Y^{8.53}A$, and $F^{8.54}A$) are on the top of H8. Since the $AT_1$ receptor acts as a model system for biased signaling, we further investigated whether these cluster mutations had effects on AngII-induced Gq activation as well as on β-arrestin 2 recruitment using BRET assays upon similar expression (Supplementary Note 20). Cluster 2 and 3 mutations completely depleted the two activation pathways of the $AT_1$ receptor, whereas cluster 1 mutations decreased both two $E_{max}$ by 22–34% (Fig. 9C, D, Supplementary Note 20). Thus, the direct perturbation at P6 influenced the G protein and β-arrestin pocket.

To further determine the potential of P6 on our intermediate state to regulate the binding of downstream transducers to the G protein pocket, we applied the AlloSigMA algorithm, which evaluates allosteric effects using the Structure-Based Statistical Mechanical Model of Allostery[70,71]. AlloSigMA analysis indicated that ligand binding on P6 influenced the G protein pocket of the receptor and that stable/bulky or unstable/tiny mutations on several P6 residues changed the dynamics of the G protein pocket (Supplementary Note 21). Thus, we designed bulky mutations ($G^{1.49}L$, $F^{7.55}W$, and $F^{8.54}W$, Fig. 10A) and tiny mutations ($F^{1.48}A$, $N^{7.49}A$, $Y^{7.53}A$, and $F^{8.50}A$, Fig. 10B) in the $AT_1$ receptor and used BRET assays to test the regulation ability of P6 on both G protein and β-arrestin pathways. Under similar protein levels (Supplementary Note 22), point mutations disturbed both Gq and β-arrestin 2 activities (Figs. 10C, D, and Supplementary Note 22). In all mutations, although the $F^{8.54}W$ mutant retained the same $E_{max}$ for Gq as the WT $AT_1$ receptor, the higher $EC_{50}$ value suggested a weaker effect of AngII. Thus, point mutations guided by AlloSigMA suggested the existence of P6.

To investigate the effects of mutations at the potential allosteric site on the AngII-induced $AT_1$ receptor conformational change, we performed intramolecular FlAsH-BRET assays using our recently developed $AT_1$ receptor conformational sensor, which

was generated by incorporation of Rluc at the C-terminus and FlAsH motif (CCPGCC) into the ICL3 (referred to as $AT_1R$ conformation sensor)[72,73]. As expected, AngII stimulation on the $AT_1R$ conformation sensor decreased the BRET signal in a dose-dependent manner, suggesting an increase of the distance between the C-terminus and the FlAsH motif inserted at the ICL3 of the $AT_1$ receptor (Fig. 10E, F). Notably, albeit with similar total and cell surface expression levels compared with the WT sensor, mutations at all seven sites led to the impairment of AngII-induced $AT_1$ receptor conformational change, as revealed by the increased $EC_{50}$, whereas $G^{1.49}L$, $F^{1.48}A$, $N^{7.49}A$, $Y^{7.53}A$, and $F^{8.50}A$ mutants also displayed decreased maximal response (Supplementary Note 22). These mutational effects on the downstream Gq signaling are consistent with the Gq BRET assay, suggesting that the $AT_1R$ conformation sensor can primarily reveal the conformational changes required for the $AT_1$ receptor Gq signal. These data suggested that the potential allosteric site participated in AT1 receptor signal regulation through conformational modulation. Hence, considering the shared existence of P6 in the class A GPCR family, it is possible to develop general allosteric modulators targeting P6 and regulating GPCR activation.

## Discussion

GPCRs have gained increasing attention as therapeutic targets for the treatment of sensory, circulation, and central nervous system disorders. Despite intensive research and the accumulation of considerable amounts of structural and functional data, it is still challenging to completely understand the mechanism of GPCR activation, elucidate the complex signaling machineries that can adopt a slew of distinct conformations to modulate multiple downstream signaling pathways (known as biased signaling), and design small-molecule allosteric modulators that can distinguish between closely related receptor subtypes[1,22]. As a quintessential model system for GPCR research, the $AT_1$ receptor not only shows biased signaling by regulating its endogenous ligand AngII, but also shows great promise as a therapeutic target for treating hypertension. Recently, the antagonist-bound (inactive), AngII-bound, and biased agonist-bound (active) $AT_1$ receptor structures have been determined[23–25]; however, the dynamic process of

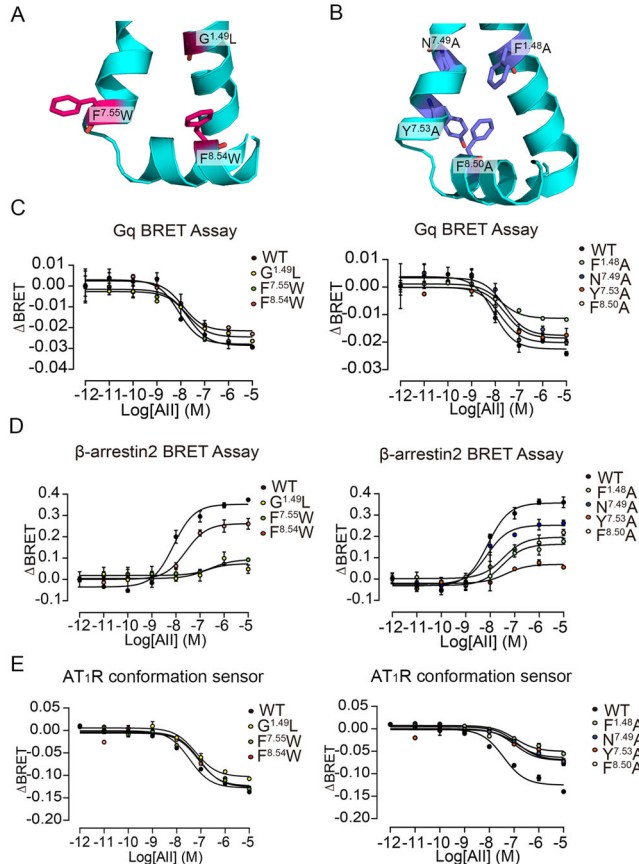

**Fig. 10 Point mutations confirm P6 as an allosteric site.** The positions of bulky (**A**, magneta) and tiny (**B**, purple) mutations around P6. AngII-induced Gq activation (**C**) and β-arrestin 2 recruitment (**D**) in HEK293 cells transfected with WT AT1 receptor or mutants. **E** AngII-induced conformational change of AT1 receptor AT1R conformation sensor in HEK293 cells transfected with AT1R conformation sensor or sensor-based mutants. In (**C–E**), points for WT, G$^{1.49}$L, F$^{7.55}$W, F$^{8.54}$W, F$^{1.48}$A, N$^{7.49}$A, Y$^{7.53}$A, and F$^{8.50}$A mutants are colored black, yellow, blue, red, cyan, dark blue, pink, and light yellow, respectively. Three independent experiments were performed, and representative dose–response curves were shown in (**C–E**). The bars indicate the mean ± SEM values. The data summary was shown in Supplementary Note 22.

receptor activation and its biased signaling through a conformational selection mechanism are still not fully understood[10,74].

In the present study, we employed extensive MD simulations (300 µs) of the AT1 receptor and used the NEB method to define a pathway from the inactive to the active state. A free-energy landscape was identified and confirmed by MSM by referring to the common GPCR activation properties. MSM extracted representative conformations of the inactive, intermediate, and active states; consequently, a synergistic activation mechanism was proposed by comparing these conformations and confirming the existence of the intermediate state. Then, tICA was applied to analyze the global movement, and it identified distinct active states, which provide hints for uncovering the biased signaling pathway mediated either by G proteins or β-arrestins. NMR experiments in future may further confirm the macrostates and the biased signaling for the AT1 receptor. Between TM7 and H8, a cryptic allosteric site was discovered in the intermediate state and confirmed by site-directed mutagenesis and biosensor. Cumulatively, our simulations and experiments make a significant

contribution to the currently available knowledge as it provides insights into the AT1 receptor activation mechanism and reports an allosteric pocket.

In general, the activation mechanism of the AT1 receptor is similar to that of other class A GPCRs. During activation, TM6 moves outward to provide space for the transducers, and TM5 and TM7 approach to form the pocket. Notably, H8 experiences an upward movement that creates a cryptic allosteric site during transition[21,48]. However, upon activation, the AT1 receptor has several other unique properties due to its particular residue composition. For instance, R$^{3.50}$ and N$^{6.30}$ show weak polar interactions rather than a strong ionic lock as observed in other GPCRs, which facilitate the conformational rearrangements of TM6 to engage with the downstream effectors and increase the proportion of the active conformations in the apo ensemble[24]. This may lead to a relatively high constitutive activity. Interestingly, the approach of Y$^{7.53}$ and Y$^{5.58}$ promotes the formation of the active state, which is unique to the AT1 receptor. Thus, the AT1 receptor not only harbors a common activation mechanism, but also maintains its uniqueness through the movement of specific residues.

Notably, intermediate states and active states exist naturally in the apo AT1 receptor. This population distribution has been proven in other apo GPCRs as means of basal activity[75–78]. Thus, agonists select and stabilize the active conformation of GPCRs and promote the binding of transducers. Once the transducers bind to the GPCRs, the active conformation harbors lower energy, hereafter the fully stabilizes the active state[52,77,79], and proves the conformational selection mechanism of GPCR activation. The active AT1 receptor showed two distinct conformations, as determined by tICA, and thus supports the hypothesis that both the Gq-bound and β-arrestin 2-bound active conformations exist in the ensemble of the AT1 receptor. The two conformers are stabilized by Gq and β-arrestin 2 biased agonists, respectively[38], suggesting a population shift mechanism in the biased signaling[80,81]. In contrast, the balanced agonists equally stabilize the two conformers, and thus lead to no preference for the downstream proteins[75]. Therefore, the emergence of biased signaling occurs via a conformational selection mechanism. In other words, the regulation of GPCR activity is attributed to a generalized allosteric modulation[82–84]; hence, it is natural to design modulators targeting allosteric sites other than endogenous orthosteric sites[85,86].

In the intermediate state, we discovered P6 as a site for allosteric regulation. Hidden between TM7 and H8, P6 was only observed during the upward movement of H8. Community analysis indicated that the impeding signal transmission between H8 and other parts impeded the formation of an active conformation (Supplementary Note 9). Mutagenesis experiments further confirmed that the residues around P6 transmit signals that modulate both G proteins and β-arrestin, and suggest that the allosteric perturbation from P6 has the potential to modulate the activity of the AT1 receptor. Several GPCRs currently lack features of endogenous ligands to initiate specific biased signaling. Thus, directly targeting intracellular TMs (such as P6) by allosteric modulators may provide a strategy to selectively regulate GPCRs[75,87]. In particular, as an unexploited allosteric site, P6 can be targeted in future allosteric drug designs for GPCRs. In order to fully realize P6 as an allosteric target for GPCRs, further structure-based drug design methodology and experimental investigation will be deployed to discover lead compounds. These studies are not only expected to provide insights into the activation mechanism of the AT1 receptor, but also offer versatile applications for GPCR biology, biophysics, and medicinal chemistry.

## Methods

**MD simulation system setup.** The inactive AT1 receptor structure in complex with an antagonist, ZD7155, and a co-crystallized apocytochrome b562RIL (PDB

4YAY), as well as the fully active structure complexed with a partial agonist, S1I8, and G protein mimetic nanobody (PDB 6DO1), were downloaded from the Protein Data Bank (PDB). Both the nanobody and agonist, as well as the antagonist and $b_{562}$RIL were removed from 6DO1 and 4YAY, respectively. In both structures, the truncated N-terminal, ECL2, and ICL3 were recovered by a loop building program, according to the sequence of the WT $AT_1$ receptor. The hydrogens were added, while the termini were capped with acetyl and methylamide groups. Referring to the common sequence in the solved structures, the 9–317 residues in 4YAY and 6DO1 were extracted as the input structures for the following processes.

**NEB sampling**. To characterize the activation process of the $AT_1$ receptor, the NEB method, a compelling approach to determine the transition pathways among conformations, was performed. During the NEB process, a series of replicas (or images) with the same atom composition but different atom coordinates are generated between the initial and final states. The replicas are linked to their nearest neighbors via elastic bands, in order to confirm that they are evenly distributed along the pathway and prevent them from directly sliding down to the energy basin[88]. Consequently, the replicas constitute a certain pathway that connects the start and end conformations.

Then, a simulated annealing process was applied to the replicas to find an MEP between the fixed initial and final states. In this process, the elastic bands were denoted as $3 \times N_{atoms}$ dimensional vectors $[\mathbf{R_0}, \mathbf{R_1}, ...., \mathbf{R_N}]$, corresponding to the pathway from the initial state $\mathbf{R_0}$ and end state $\mathbf{R_N}$. To avoid interference of spring forces caused by the restraints and potential forces caused by the force fields, a tangent vector $\tau_i$ at each replica position was introduced. With respect to the energy of replica i ($V_i$) and its neighbors, $\tau_i$ is defined in Eq. (1).

$$\tau_i = \begin{cases} \mathbf{R_{i+1}} - \mathbf{R_i}, & V_{i+1} > V_i > V_{i-1} \\ \mathbf{R_i} - \mathbf{R_{i-1}}, & V_{i+1} < V_i < V_{i-1} \\ (\mathbf{R_{i+1}} - \mathbf{R_i})\Delta V_i^{max} + (\mathbf{R_i} - \mathbf{R_{i-1}})\Delta V_i^{min}, & V_i > V_{i+1} > V_{i-1} \text{ or } V_{i+1} > V_{i-1} > V_i \\ (\mathbf{R_{i+1}} - \mathbf{R_i})\Delta V_i^{min} + (\mathbf{R_i} - \mathbf{R_{i-1}})\Delta V_i^{max}, & V_i > V_{i-1} > V_{i+1} \text{ or } V_{i-1} > V_{i+1} > V_i \end{cases} \quad (1)$$

where $\Delta V_i^{max} = \max(|V_{i+1} - V_i|), (|V_{i-1} - V_i|)$, $\Delta V_i^{min} = \min(|V_{i+1} - V_i|), (|V_{i-1} - V_i|)$

With $\tau_i$, Eqs. (2) and (3) define the perpendicular ($\mathbf{F_i^\perp}$) and parallel ($\mathbf{F_i^\parallel}$) components of the total force, respectively.

$$\mathbf{F_i^\perp} = -\nabla V(Pi) + (\nabla V(P_i) \cdot \tau_i) \cdot \tau_i \quad (2)$$

$$\mathbf{F_i^\parallel} = (\mathbf{F^s} \cdot \tau_i) \cdot \tau_i \quad (3)$$

$$\mathbf{F^{total}} = \mathbf{F_i^\perp} + \mathbf{F_i^\parallel} \quad (4)$$

where $\nabla V(P_i)$ represents the gradient of the energy regarding the atomic coordinates in replica $i$, namely the potential force originating from the force fields. $\mathbf{F^s}$ is the force provided by the elastic bands. As a result, the force field parameters only contribute to the perpendicular part of the total force, and the spring force is only responsible for the parallel part [Eq. (4)][89,90].

After the preparation, the inactive (PDB 4YAY) and active (PDB 6DO1) structures were set as the initial and end states, respectively. The Amber ff14SB force field was adapted for atom interactions[91]. First, we conducted 4000 steepest descent and 6000 conjugate gradient energy minimization cycles to our two systems. Then, 20 replicas were created between the inactive and active $AT_1$ receptors with elastic bands connecting each other. The translational and rotational differences between the replicas were excluded by moving the systems to the coordinate of the center of mass and applying an optimal rotation matrix to the replicas to minimize the RMSD between the systems. In the initial heating process, the systems were gradually heated from 0 to 300 K with a spring force of 10 kcal mol$^{-1}$ Å$^{-2}$, a timestep of 0.5 fs, and a Langevin collision frequency of 1000 ps$^{-1}$. During the next equilibration, simulated annealing, and cooling runs, a spring force of 50 kcal mol$^{-1}$ Å$^{-2}$ was used. The equilibration runs of the replicas were carried out at 300 K with a timestep of 1 fs. In the simulated annealing runs, the systems were first heated to 500 K, then gradually cooled and equilibrated alternatively at 0 K with a timestep of 0.5 fs. Finally, the replicas were completely cooled at 0 K for 2 ns with a timestep of 1 fs. The detailed workflow has been confirmed in the previous studies[27,92].

**MD simulation settings**. The NEB process provided 22 replicas, but some had highly similar conformations. Thus, we calculated the RMSD between adjacent structures and accordingly selected the most different 15 replicas, including the starting and end structures, as the initial structures for the MD simulations.

All structures were oriented in the Orientations of Proteins in Membranes server[93]. Next, they were inserted into a POPC membrane using the CHARMM36 additive force field in the CHARMM-GUI server[94]. TIP3P water molecules with a length of 10 Å were added to the top and bottom of the system. The counterions K$^+$ or Cl$^-$ and an additional 0.15 mol L$^{-1}$ KCl were also solvated in the systems. The components of bilayers have been commonly used in other simulations[38,41]. With the help of the input generator of CHARMM-GUI, we obtained the Amber format coordinate and topology files.

The systems were first minimized with restraint of 500 kcal mol$^{-1}$ Å$^{-2}$ on the $AT_1$ receptor and lipids, while water and counterions were minimized in 8000 steepest descent cycles, followed by 7000 conjugate gradient cycles. Second, all atoms encountered $1.5 \times 10^4$ cycles of steepest descent and $1.5 \times 10^4$ cycles of conjugate gradient minimization. Next, with 10 kcal mol$^{-1}$ Å$^{-2}$ position restraint on proteins and lipids, the systems were gradually heated from 0 to 300 K in 300 ps and equilibrated for 700 ps under canonical ensemble conditions. Then, the 15 systems underwent 10 rounds of 2 µs MD simulations, with an integration step of 2.0 fs. Finally, we collected 150 independent repeat trajectories with random initial velocities. The total simulation timescale was 300 µs. During simulations, the Particle mesh Ewald method was applied to calculate the long-range electrostatic interactions, and a cutoff of 10 Å was used for short-range electrostatic and van der Waals interactions. The SHAKE algorithm was employed for covalent bonds containing hydrogen. A temperature of 300 K was controlled by Langevin Thermostat, while the collision frequency was 1.0 ps$^{-1}$. The snapshots were written out every 200 ps.

**Markov State Model construction**. According to the activation parameters, an MSM was built using the PyEMMA protocol (http://www.emma-project.org/latest/)[95]. Through the implied timescale validation (Supplementary Note 5), we confirmed that the $AT_1$ receptor systems were Markovian and reliable with a 200 microstate model with a lag time of 8 ns and a maximum k-means iteration number of 200. Then, the microstates were clustered into three macrostates via the PCCA+ algorithm, which was confirmed by a Chapman–Kolmogorov test. Using TPT, we measured the transition probability matrix of the MSMs and computed the mean first passage time for each activation and inactivation process[96]. Based on the *mdtraj* package, we extracted the structures close to the microstate cluster centers of each macrostate into the trajectories for the corresponding macrostates. Then, using three trajectories, we selected the representative conformation of each macrostate according to the similarity score $S_{ij}$. As shown in Eq. (5), the conformation with the highest $S_{ij}$ among the trajectories was regarded as the most representative conformation of the macrostate. The $d_{ij}$ is the RMSD between the conformations i and j, while $d_{scale}$ is the standard deviation of $d$.

$$S_{ij} = e^{-d_{ij}/d_{scale}} \quad (5)$$

**Measurement of receptor expression by ELISA**. HEK293 cells were transiently transfected with Flag-tagged WT $AT_1$ receptor or mutants, Flag-tagged V2R receptor, or Flag-tagged $AT_1$R conformational sensor or sensor-based mutants in 24-well plates. After incubation at 37 °C for 48 h, the cells were fixed with DPBS containing 4% (w/v) formaldehyde for 5 min at room temperature. For whole-cell ELISA, the cells were incubated in blocking solution (5% BSA in DPBS) containing 0.2% Triton X-100 for 1 h at room temperature. For cell surface ELISA, the cells were incubated in blocking solution without Triton X-100. The cells were washed three times with DPBS and incubated overnight with an anti-Flag primary antibody (Sigma Aldrich, Cat# F1804, 1:1000) at 4 °C, followed by incubation with a horseradish peroxidase-conjugated secondary anti-rabbit antibody (Thermo Fisher, Cat# A-21235, 1:5000) for 1 h at room temperature. After washing, tetramethyl benzidine solution was added and the reaction was stopped by adding an equal volume of 0.25 M HCl solution. The solution was incubated for 5 min. The optical density of each well was measured at 450 nm using the TECAN luminescence counter (Infinite M200 Pro NanoQuant). The optical density was plotted against the transfecting amounts of respective plasmids to determine the relative expression levels of each receptor or mutants. The levels of the $AT_1$ receptor mutants were normalized to that of the WT receptor.

**Site-directed mutagenesis**. All $AT_1$ receptor mutants used in the present study were generated by site-directed mutagenesis. The successful introduction of the mutations in the polymerase chain reaction products was verified by DNA sequencing. Corresponding primer sequences were shown in Supplementary Data 2.

**BRET measurement**

*β-arrestin 2 recruitment*. HEK293 cells were transiently cotransfected with β-arrestin 2-Rluc and YFP-tagged WT or mutated $AT_1$ receptor. After 24 h, the cells were seeded on 96-well microplates and incubated for 24 h at 37 °C, after which were washed twice with HBSS and stimulated with AngII at different concentrations. Luciferase substrate coelenterazine-h (Promega) was added at the concentration of 5 µM before light emissions were recorded using a Mithras LB940 microplate reader (Berthold Technologies). The BRET signal was determined by calculating the ratio of luminescence at 530/485 nm.

*G protein activation*. Gq (Gq-RLuc8, Gβ3, Gγ9-GFP10), Gi (Gi1-RLuc8, Gβ3, Gγ9-GFP10), and G12(G12-RLuc8, Gβ3, Gγ9-GFP10) BRET probes were from the TRUPATH kit, which was a gift from Bryan Roth (Addgene kit #1000000163)[97]. HEK293 cells were transiently co-transfected with WT or mutated $AT_1$ receptor along with specific G protein BRET probes according to the experimental setting. After 24 h, the cells were seeded on 96-well microplates and incubated for an additional 24 h. For the constitutive activity measurement, cells transfected with varying amounts of WT or mutated $AT_1$ receptor, or V$_2$R were washed twice with

HBSS buffer and the BRET signal was directly recorded after the addition of 5 μM luciferase substrate coelenterazine 400a using a Mithras LB940 microplate reader. For the AngII-stimulated G protein activation, the cells were washed twice with HBSS and stimulated with AngII at different concentrations. BRET signal was subsequently measured after the addition of the luciferase substrate and was calculated as the ratio of light emission at 510/400 nm.

The data obtained in the G protein activation assay and β-arrestin recruitment assay were normalized as a percentage of the $E_{max}$ of WT AT$_1$ receptor (reference receptor) in each pathway. The normalized data were analyzed using the "Operational Model" in GraphPad Prism to determine the transduction coefficient $\log(\tau/K_A)$, in which $\tau$ is the transducer ratio and $K_A$ is an agonist-receptor dissociation constant[98], of the WT AT1 receptor and mutants at each signaling pathway[99,100].

The equation for the "Operational Model" was defined by the following parameters:

Equation Type - Explicit Equation: Y = a function of X and parameters
Definition: $A = 10^X$
operate1 = $((1 + A)/(10^{LogR}*A))^n$
operate2 = $((1 + A/(10^{LogK_A}))/((10^{LogR}*A))^n$
Y1 = basal + (Emax − basal)/(1 + operate1)
Y2 = basal + (Emax − basal)/(1 + operate2)
<A:O>Y = Y1
<~A:O>Y = Y2

The relative activities [$\Delta\log(\tau/K_A)$] of AT$_1$ receptor mutants in each signaling pathway was determined by the Eq. (6):

$$\Delta\log\left(\frac{\tau}{K_A}\right) = \log\left(\frac{\tau}{K_A}\right)_{mutant} - \log\left(\frac{\tau}{K_A}\right)_{WT} \quad (6)$$

The SEM of the $\Delta\log(\tau/K_A)$ was calculated according to Eq. (7):

$$SEM_{\Delta\log(\frac{\tau}{K_A})} = \sqrt{(SEM_{mutant})^2 + (SEM_{WT})^2} \quad (7)$$

*Intramolecular FlAsH-BRET assay.* The AT$_1$R conformation sensor was generated by fusing Rluc to the C-terminus of the AT$_1$ receptor and inserting a TC-tag (CCPGCC) between K224 and A225 at ICL3. HEK293 cells were transfected with the AT$_1$R conformation sensor without or with different mutations. After 48 h, the cells were labeled with a 2.5-μM FlAsH-ETD2 from a TC-FlAsH II In-Cell Tetracysteine Tag Detection Kit (Thermo Fisher Scientific) according to the manufacturer's instructions. The cells were then stimulated with varying concentrations of AngII and BRET signal was measured after addition of 5 μM RLuc substrate coelenterazine-h using a Mithras LB940 microplate reader. The BRET signal was determined as the ratio of luminescence at 530/485 nm.

**Reporting summary.** Further information on research design is available in the Nature Research Reporting Summary linked to this article.

## Data availability

The data that support this study are available from the corresponding authors upon reasonable request. The activation parameter and tICA data generated in this study are provided in the Source Data file. Initial structures for MD simulation were obtained from the RCSB PDB database (PDB 4YAY and 6DO1). Other GPCRs were also downloaded from the RCSB PDB database (https://www.rcsb.org/). The analysis protocol for Markov State Model referred to http://www.emma-project.org/latest/. The network analysis in Supplementary Note 9 was provided by http://www.scs.illinois.edu/schulten/tutorials/network/. Pocket prediction was accomplished by fpocket, see http://fpocket.sourceforge.net/. Other simulation analyses were based on AMBER suite, according to http://ambermd.org/. Source data are provided with this paper.

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

## Acknowledgements

We thank the support from the AlloStar™ platform of Nutshell Therapeutics. This work was partly supported by grants from the National Natural Science Foundation of China (81925034, 22077082, 21778037, 91753117, 81721004, and 91939301), the Innovation Program of Shanghai Municipal Education Commission (2019-01-07-00-01-E00036, China), the

Shanghai Science and Technology Innovation (19431901600, China), and the Shanghai Health and Family Planning System Excellent Subject Leader and Excellent Young Medical Talents Training Program (2018BR12, China), CAMS Innovation Fund for Medical Sciences (CIFMS) (2019-I2M-5-051), the Fundamental Research Funds for the Central Universities, and the Center for high Performance Computing and System Simulation, Pilot National Laboratory for Marine Science and Technology (Qingdao).

## Author contributions

S.L. and J.Z. conceived and supervised the project. S.L., J.Z., J.S., and X.H. designed the experiments and revised the manuscript. X.H., Z.Y. and S.Z. designed and performed the experiments, and drafted the manuscript. X.H. contributed to MD simulations and data analysis. Z.Y., S.Z., and J.W. acquired and analyzed BRET data. Z.C., A.R., D.N., and J.P. acquired data and revised the manuscript. S.L., J.Z., and J.S. were responsible for the conception and oversight of the project. All authors discussed the results and reviewed the manuscript.

## Competing interests

The authors declare no competing interests.

## Additional information

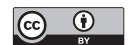

