## [Peer Review File · Nature Communications]

Reviewers' Comments:

Reviewer #1:

Remarks to the Author:

Comments to

He et al 2020 Nature Communications

This study employs advanced and extended MD simulations, nudged elastic band method and Markov state models as well as additional, complementary in silico analyses to define an activation path of the angiotensin II type 1 receptor. While the authors provide an impressive amount of MD combined with clever complementary in silico approaches the manuscript suffers from a poorly defined terminology regarding the inactive, intermediate and active state of the receptor (the fully active state should be defined as a ternary complex of agonist, receptor and intracellular binder such as G protein/mini G/nanobody). While the data are based on existing receptor structures as starting points, no additional attempts are done to validate the in silico findings in wet lab experiments addressing structure or function. The provided analysis is indeed exciting providing a glimpse into receptor transition states and potential conclusions. However, this reviewer's excitement is reduced by a lack of clarity. Also, the authors should be humble about firm statements presented as solid findings, where they rather pose a novel hypothesis.

1. The authors do not provide the information whether the nanobody is removed from the extended MD in the PDB6DO1. That information is crucial.

2. INTRODUCTION:

a. Authors describe the GPCR activation process as ligand-induced. They should not neglect the possibility of constitutive activity leading to receptor activation in a ligand-independent manner. Especially since the whole in silico analysis is done in the absence of ligands, as I understand it.

b. Orthosteric sites of GPCRs are described to be located "within the 7TM helices" and that statement is wrong if it is not further clarified.

c. The recent advance in solving receptor structures is not a result of receptor and protein engineering but rather technological breakthroughs e.g. in the area of CryoEM or the development of X-ray free electron lasers.

d. In the description of which receptor structures are active and inactive the authors are severely inaccurate, which also has bearing for one of my major points mentioned below. The authors describe antagonist-bound GPCRs as inactive, which is most likely correct, even though an inverse agonist-bound structure is – most likely – less active, given the fact that an antagonist has no efficacy. Furthermore, the authors describe the agonist-bound structure as active and that is in part incorrect or incomplete. In this context, the authors must mention the necessity of an allosteric modulator in form of a G protein, mini G protein, arrestin or nanobody to stabilize a fully active receptor.

e. This reviewer agrees that most experimental approaches cannot reveal the conformational landscapes of GPCRs with similar atomic resolution, but conformational biosensors and NMR studies DO allow to for the investigation of intermediate states. Check for instance <https://doi.org/10.1073/pnas.1900261116>; DOI: 10.1126/science.1215802

f. Towards the end of the introduction, the authors touch upon activation paths and metastable states. Here, it might be suitable to refer to rhodopsin, where these issues have been dissected in more detail. Especially since plenty of rhodopsin structures are listed in Table S1.

g. Page 4 lane 97/98 – again, the PDB-listed structures should be better defined by mentioning the presence of the nanobody required to stabilize the fully active receptor (PDB 6DO1).

h. Lanes 109-112 – I feel that that has been stated before, appears repetitive.

i. Lane 113/114 – NO! The fact that there are no allosteric modulators for a receptor reported does by no means suggest that allosteric sites do not exist! This statement appears like a very weak justification for the aim of the current study.

j. IUPHAR nomenclature for dopamine D3 receptors includes numbers in subscript.

3. RESULTS

- a. The PDB 6DO1 must be described as nanobody-and agonist-bound active structure. Information if the nanobody was removed for MD must be included here and in the materials and method section.
- b. Justify, why solely POPC was used and not POPC/cholesterol, which could have an allosteric modulatory role.
- c. Page 5, lanes 149-150: What is the rationale that the authors base their analysis on the RELATIVE movement of TM5 to TM7? To define the outward movement of TM6, the authors refer to a more rigid part of the receptor (i.e. TM2), so why don't they do that for TM5 and TM7 as well? Wouldn't it be wiser to use the same approach to assess the movements of TM5, 6 and 7?
- d. Fig. 2: The projection of previously obtained structures generally confirms the approach of the authors. Though, special attention should be given to the significant outliers in Fig. 2B (e.g. the blue, inactive structure that is almost in the active-state cloud). Are these outliers "exotic" GPCRs that are known to undergo a rather distinct activation mechanism?
- e. The key reference 21 does not report any MD.
- f. The authors introduce the topic of receptor activation arguing that it is ligand-driven (see my comment above). The inactive structure that is the starting point of the NEB and MD is antagonist bound, whereas the defined endpoint refers to an agonist- and nanobody bound active structure. However, the authors do not mention if either the antagonist has been removed for the MD or if the agonist was introduced in the inactive structure to facilitate the activation process. If ligands are not present, the authors observe constitutive activity in their MD; this must be clarified. The authors should provide experimental (wet lab) data providing the notion of AT1R constitutive activity. This is not only to confirm their findings from the in-silico experiments, but it would also indicate which signaling pathways this active receptor state reflects (AT1R can couple to multiple G protein isoforms and b-arrestin). At least a paper should be cited, where constitutive activity of AT1 receptor was investigated.
- g. Supplementary table Section 2 S1: I just quickly looked at this eminent summary of GPCR structures and was surprised to find only one AT1 receptor structure. Lacking in the table: 4YAY, 5UNG, 6OS0, 6OS1, 6OS2. Are those structures included in the data in Fig. 2B? Did not check the rest of the table.
- h. In Section 2. What is meant by "the state of the protein referred to the gpcrib database"? No column in the Table S1 has the title "state of protein" – do the authors refer to activity? Also, given the rapid pace of the appearance the authors should state a date of the table, especially since they claim comprehensiveness.
- i. In the context of defining active state, the authors need to provide the information if "active" means agonist and G protein (mG, nanobody)-bound or only agonist-bound receptor in Table S1. As mentioned above, an agonist-bound structure is only fully active in the presence of the G protein (or mG, arrestin, nanobody)
- j. Out of curiosity: are the arrestin structures clustered and are they clustered at a different region compared to G protein-bound structures? Where are nanobody-bound structures (do nanobody structures sample both arrestin and G protein cluster?).
- k. Page 6 – second paragraph, what about fully inactive, inverse agonist-bound structures?
- l. On page 8 the authors finally mention that the analysis was done in the apo state. Thus, the authors observe constitutive rather than agonist-induced activation. This must be stated clearly. On the other hand, this might offer space for control experiments. Could the authors employ an intermediate state and add agonist/inv agonist and follow activation towards more active and inactivation towards inactive as defined in Fig. 3?
- m. Page 9, lanes 261 downward: Is the ECL2 closure of the orthosteric site driven by the G protein mimicking nanobody as reported in the allosteric coupling paper from DeVree et al., Nature, 2016? Why does the upward movement of Helix 8 open a ligand binding pocket? The authors should provide a better rationale (maybe with additional figures) why they are concluding this here. And most importantly: AT1 receptors can activate multiple G protein subtype upon agonist binding (G12/13, Gi/o, Gq). Is the transition of AT1R to the active state through this proposed intermediate state required for the activation process of all signaling pathways? A combination of in-silico and in-vitro experiments is needed to confirm this core statement of the manuscript. (Suggestion for in-vitro

experiments: G protein sensor activation upon stimulation of AT1R mutants where the key microswitches required for the intermediate state are abolished).

n. In the large table S1 it is instrumental (as mentioned above) to be absolutely clear about what is meant with "active", "inactive" and "intermediate" receptor. For that purpose, the authors must list presence of e.g. G protein, nanobody, mini G protein, arrestin etc. The referral to the gpcrdb interpretation of activity is not sufficient. I only went in to some examples and found e.g. EP3 (PDB6AK3), a receptor that is listed with active activity in the absence of G protein or nanobody, whereas a g4LDO (beta2) is equally listed as active but has a nanobody bound. I fear that this poorly defined handling of receptor activity compromises the analysis and especially the MSM presented in Fig. 3. How does the MSM handle the difference in active state given presence and absence of G protein? The authors discuss the relevance of constitutive activity shortly on page 9, but in the remaining text, the differentiation between basal activity vs agonist-induced and G protein-stabilized activity remain obscure.

o. Page 11: Referral to ref 27 and the "canonical active" conformation adds to the confusion of what the authors define as "active". (btw the authors should consider the use of the term "canonical" – in science history it has happened that findings that appear canonical one day aren't anymore the next day). Maybe that should be replaced by G protein-bound or something in that direction.

p. Page 12: Modelling of active conformations with G protein and arrestin. These in silico experiments need to be explained better. Since the macrostates were developed in the absence of G protein/arrestin the modeling of the complex appears to this reviewer as a weak argument for bias. Experimental evidence from in-vitro studies is needed to support this finding. NMR studies with purified and labeled AT1R would probably allow for the most direct validation of these proposed macrostates but also signaling (=activation) readouts with AT1R mutants could further underline the authors' claims. The authors neglect the fact that AT1R can also couple to G proteins from the G12/13 and Gi/o family. Given that they claim at least 8 macrostates of this receptor (four of them being at least partially active), it is essential to also assess (computationally and experimentally) their effects on other signaling pathways than Gq and beta-arrestin.

q. Discovery of a cryptic allosteric site (please add references to the approach Fpocket). Four of the identified pockets (P2, P3, P4 and P5) are localized on the outer surface of the receptor. How can this be defined a binding "pocket"? The authors should elaborate more on the methodological details of their Fpocket approach. The statement that the authors discovered a new, likely allosteric site appears weak, without an actual proof to target this site with a small molecule. At least a docking analysis could be provided (even though I understand that this is out of the scope of the paper). It appears rather like a well-founded hypothesis that requires testing. In Fig 9A, the authors could highlight the residues that are creating the hypothesized P6 allosteric site for clarity. The mutational alanine mutation appears like a very blunt tool to argue for the existence of a cryptic allosteric site in a mutational cluster. Are all these 5 clusters really involved in the formation of P6? From the structures in Fig. 8 and Fig. 9, I would think that it is only one TM (TM1 or TM2?) + TM7 + H8, meaning only three clusters should be examined. The authors do not explain why they chose these 5 clusters and therefore the correlation between their signaling data and P6 formation is substantially hampered! While the authors claim that cluster mutation underline the hypothesis of an allosteric P6 ligand binding pocket, I wonder whether this is this because a) P6 is not a "pocket" but rather a microswitch or b) because an endogenous allosteric modulator binds to P6 in wildtype but not mutant AT1R?

r. Regarding the functional readout of G protein activation and arrestin recruitment. The G protein readout is based on a decrease in BRET as a result of Gq dissociation (TRUPATH sensors). The arrestin recruitment is based on a direct BRET between YFP-tagged receptors and Rluc-tagged arrestin. While the authors argue about receptor conformations in this manuscript, none of these assays probes for receptor conformation (the arrestin could be a matter of discussion – I agree). I would suggest to rather use BRET probes that act as sensors of receptor conformation, such as a tagged nanobody or a tagged mini G protein. In the figure legend, no information about experimental repetitions, curve fitting, normalization is provided.

Minor comments:

1. The official nomenclature of the angiotensin II type 1 receptor according to IUPHAR is AT1 receptor

(number in subscript) – this should be used throughout the work.

2. 1st sentence of the abstract: receptors are described as frequent targets – this feels awkward, rather “are the most frequently targeted...”

3. G protein rather than G-protein

Reviewer #2:

Remarks to the Author:

In the manuscript titled “Activation pathway of a G protein-coupled receptor uncovers conformational intermediates as novel targets for allosteric drug design”, the authors used AT1R as the prototype GPCR, and used a computational framework including transition pathway generation program NEB and simulated annealing to generate 15 structures on the MEP, and followed by 10 2 μ s- unbiased MD runs for each structures (totally 300 μ s) to investigate the landscape of its dynamic activation pathway. For the MSM analysis of the comprehensive simulations, they discovered canonical (Gq/arrestin balanced) and alternative (arrestin biased) active states, which is in consistent with the recently reported biased agonism study of AT1R by Ron Dror et al (Science, 2020, 367, 881-887). In conformational intermediates, the authors identified several allosteric pockets and reveals P6 as the novel cryptic allosteric binding site for AT1R (although it exists in several other GPCRs’ intermediates) that could be used to develop allosteric modulators of AT1R. Furthermore, they validated the P6 site through clustered alanine scanning and BRET.

In summary, this is a compelling study of AT1R’s landscape of dynamic activation pathway. I have the following minor concerns:

1. In line 148 and 150, the N7.46 seems should be N7.49.
2. In Figure 4D, the residue L6.30 should be N6.30.
3. In line 247, Y5.53 should be Y5.58.
4. In line 245, please rationalize the chosen of CD atom of R3.50. Why not CZ of R3.50?
5. In Figure 4A-C, the weak interactions (such as hbonds) and distances mentioned in the text better to be labeled.
6. In Figure 4E, the area of triangle composed of L3.43, V6.41, and I6.40 is calculated. Please rationalize the chosen of CG atom of L3.43, CB atom of V6.41 and CB atom of I6.40. Why not using CB atoms for all of them?
7. How did you design the five clustered mutations in the P6 site? What not just using point mutations?

Reviewer #1 (Remarks to the Author):

This study employs advanced and extended MD simulations, nudged elastic band method and Markov state models as well as additional, complementary *in silico* analyses to define an activation path of the angiotensin II type 1 receptor. While the authors provide an impressive amount of MD combined with clever complementary *in silico* approaches the manuscript suffers from a poorly defined terminology regarding the inactive, intermediate and active state of the receptor (the fully active state should be defined as a ternary complex of agonist, receptor and intracellular binder such as G protein/mini G/nanobody). While the data are based on existing receptor structures as starting points, no additional attempts are done to validate the *in silico* findings in wet lab experiments addressing structure or function. The provided analysis is indeed exciting providing a glimpse into receptor transition states and potential conclusions. However, this reviewer's excitement is reduced by a lack of clarity. Also, the authors should be humble about firm statements presented as solid findings, where they rather pose a novel hypothesis.

RESPONSE: We heartfully appreciate the time that the reviewer has dedicated to providing insightful suggestions on ways to improve our manuscript. According to the reviewer's comments, we have revised the definition of GPCR conformational states and performed additional experiments to validate *in silico* findings. We hope our detailed responses address your comments.

1. The authors do not provide the information whether the nanobody is removed from the extended MD in the PDB6DO1. That information is crucial.

RESPONSE: The nanobody in the active AT₁ receptor structure (PDB ID: 6DO1) was removed in the MD simulations. We have added this information in both Results (page 5) and Methods (page 30) in the revised main text.

2. INTRODUCTION:

a. Authors describe the GPCR activation process as ligand-induced. They should not neglect the possibility of constitutive activity leading to receptor activation in a ligand-independent manner. Especially since the whole in silico analysis is done in the absence of ligands, as I understand it.

RESPONSE: We have added the description of constitutive activity at the beginning of the Introduction as follows (page 3):

In addition, many GPCRs are able to transmit signals in the absence of an external stimulus or an agonist, through ‘basal’ (also known as ‘constitutive’) activity (Vecchio et al., *J. Pharmacol. Exp. Ther.* **2016**, 357, 36–44; Zhao & Furness, *Biochem. Pharmacol.* **2019**, 170, 113647; Zhang et al., *eLife*. **2018**, 7:e33432; Hu et al., *J Biol Chem.* **2014**, 289:24215-25).

b. Orthosteric sites of GPCRs are described to be located “within the 7TM helices” and that statement is wrong if it is not further clarified.

RESPONSE: Thanks for your correction. We have revised the sentence in the main text as follows (page 3):

The GPCR-mediated signal transduction is often triggered by an interaction between extracellular signal and the ligand binding site compose of the extracellular region and the 7TMs bundles...

c. The recent advance in solving receptor structures is not a result of receptor and protein engineering but rather technological breakthroughs e g in the area of CryoEM or the development of X-ray free electron lasers.

RESPONSE: We have corrected it in the revised main text as follows (page 3):

Recent technological breakthroughs in structural biology, such as cryo-electron microscopy (cryo-EM) or X-ray free electron lasers, have led to the identification an increasing number of solved GPCR structures (Josephs et al., *Science* **2021**, 37, eabf7258; Kang et al., *Nature* **2015**, 523, 561–567; Thal et al., *Curr. Opin. Struct. Biol.* **2018**, 51, 28–34).

d. In the description of which receptor structures are active and inactive the authors are severely inaccurate, which also has bearing for one of my major points mentioned below. The authors describe antagonist-bound GPCRs as inactive, which is most likely correct, even though an inverse agonist-bound structure is – most likely – less active, given the fact that an antagonist has no efficacy. Furthermore, the authors describe the agonist-bound structure as active and that is in part incorrect or incomplete. In this context, the authors must mention the necessity of an allosteric modulator in form of a G protein, mini G protein, arrestin or nanobody to stabilize a fully active receptor.

RESPONSE: We agree with your suggestion and we have re-defined the receptor conformational states in the revised main text (Latorraca et al., *Chem. Rev.* **2016**, 117, 139–155; Manglik et al., *Cell* **2015**, 161, 1101–1111; Nygaard et al., *Cell* **2013**, 152, 532–542) (pages 5 and 8).

We defined inverse agonist- or antagonist-bound forms as inactive conformations, only agonist-bound forms as active conformations, and both agonist- and G protein-, β -arrestin, or nanobody-bound forms as selective conformational states to couple downstream effectors.

e. This reviewer agrees that most experimental approaches cannot reveal the conformational landscapes of GPCRs with similar atomic resolution, but conformational biosensors and NMR studies DO allow to for the investigation of intermediate states. Check for instance <https://doi.org/10.1073/pnas.1900261116>; DOI: 10.1126/science.1215802

RESPONSE: Thanks for your suggestion. Biosensors and NMR can capture conformational changes of receptors and permit the observation of intermediate states (Grushevskiy et al., *PNAS* **2019**, 116, 10150–10155; Liu et al., *Science* **2012**, 335, 1106–1110). As you suggested, we have added the use of biosensors and NMR methods in the activation pathway exploration of receptors in the Introduction (page 4) as follows:

To uncover the activation pathway of GPCRs, biosensors, nuclear magnetic resonance (NMR), and computational methods have been widely applied (Grushevskiy et al., *PNAS* **2019**, 116, 10150–10155; Guros et al., *PNAS* **2020**, 117, 405–414; Liang et al., *Mol. Cell* **2020**, 77, 656–668; Liang et al., *ACS Pharmacol. Transl. Sci.* **2020**, 3, 263–284; Liu et al., *Science* **2012**, 335, 1106–1110; Yang, et al., *Nat Commun* **2015**, 6:8202; Yang, et al., *Nat Chem Biol* **2018**, 14:876–886; Liu, et al., *Nat Commun* **2020**, 11:4857; He, et al., *Nat Commun* **2021**, 12:2396). Among these approaches, molecular dynamics (MD) simulations have become a well-established technique for probing the conformational landscapes at an atomic level and directly uncovering biomolecular mechanisms (Hollingsworth et al., *Nat. Commun.* **2019**, 10, 3289; Suomivuori et al., *Science* **2020**, 367, 881–887; Zhang et al., *Chem. Sci.* **2019**, 10, 3671–3680).

f. Towards the end of the introduction, the authors touch upon activation paths and metastable states. Here, it might be suitable to refer to rhodopsin, where these issues have been dissected in more detail. Especially since plenty of rhodopsin structures are listed in Table S1.

RESPONSE: Thanks for your suggestion. The activation pathway of rhodopsin has been elucidated by both experimental and computational methods. Considering its important status and the availability of structures, we have added the description of the activation pathway of rhodopsin elucidated by experiments and simulations in the revised main text as follows (page 5):

In the best structurally and biochemically characterized GPCRs, the rhodopsin receptor, the activation pathway and the corresponding intermediate states have been elucidated by NMR (Patel et al., *PNAS* **2004**, 101, 10048–10053), Fourier transform infrared spectroscopy (Zaitseva et al., *JACS* **2010**, 13, 4815–4821), and MD simulations (Laricheva et al., *JACS* **2013**, 135, 10906–10909).

g. Page 4 lane 97/98 – again, the PDB-listed structures should be better defined by mentioning the presence of the nanobody required to stabilize the fully active receptor (PDB 6DO1).

RESPONSE: We have stated in revised main text that the nanobody in the 6DO1 structure is required for the stabilization of the fully active AT₁ receptor as follows (page 4):

By comparing the structures of its inactive, antagonist-bound state (ZD7155; PDB ID: 4YAY) (Fig. 1A) and the active state, meaning the AT₁ receptor complexed with a partial agonist S118 peptide, and a G protein mimetic nanobody to maintain the fully active conformation (PDB ID: 6DO1) (Fig. 1B).

h. Lanes 109-112 – I feel that that has been stated before, appears repetitive.

RESPONSE: We have rephrased the sentences to express our different emphasis.

In the original manuscript, we described the general challenge for elucidating GPCR activation pathway using static snapshots of structures in the second paragraph. In the third paragraph, we stated that the activation pathway of the AT₁ receptor is unclear and how the conformational transition landscape looks like.

We underscored GPCRs in the paragraph 2 (page 3) and AT₁ receptor in the paragraph 3 (page 4) in order to connect between them and to elicit the purpose of the study of AT₁ receptor activation.

i. Lane 113/114 – NO! The fact that there are no allosteric modulators for a receptor reported does by no means suggest that allosteric sites do not exist! This statement appears like a very weak justification for the aim of the current study.

RESPONSE: We agree with your comment that no report of allosteric modulators does not mean no allosteric pocket exists. In the revised main text, we have rephrased this sentence as follows (page 4):

Furthermore, despite the availability of the inactive and active structures of the AT₁ receptor, there are no allosteric modulators of this receptor reported to date, suggesting a challenge for targeting potential allosteric binding sites in the two

available snapshots. However, a cryptic allosteric site may exist in the transition pathway.

j. IUPHAR nomenclature for dopamine D₃ receptors includes numbers in subscript.

RESPONSE: We have changed it to dopamine D₃ receptors in the main text (page 4). In addition, all GPCR names in our manuscript have been revised referring to IUPHAR nomenclature.

3. RESULTS

a. The PDB 6DO1 must be described as nanobody-and agonist-bound active structure. Information if the nanobody was removed for MD must be included here and in the materials and method section.

RESPONSE: We have stated that the nanobody in PDB 6DO1 was removed before MD simulations in both Results (page 5) and Methods (page 30) and defined both agonist- and nanobody-bound forms as a fully active conformation (PDB ID 6DO1).

b. Justify, why solely POPC was used and not POPC/cholesterol, which could have an allosteric modulatory role.

RESPONSE: In MD simulations, cholesterol-rich bilayers were applied to explore the allosteric effect of cholesterol in particular (Sengupta & Chattopadhyay, *Biochim. Biophys. Acta* **2015**, 1848, 1775–1782), but our purpose was not to elucidate the allosteric effect of cholesterol in the AT₁ receptor. Cholesterol can stabilize GPCRs in certain conformation, leading to block population shift of the receptors (Manna et al., *Elife* **2016**, 5, e18432). In addition, pure POPC membrane is a commonly-used GPCR environment during MD simulations, which have provided considerable important conformational changes of the receptors (Dror et al., *PNAS* **2011**, 108, 18684–18689; Kohlhoff et al., *Nat. Chem.* **2014**, 6, 15–21; Mattedi et al., *PNAS* **2020**, 117, 15414–15422; Suomivuori et al., *Science* **2020**, 367, 881–887; Hollingsworth et al., *Nat. Commun.* **2019**, 10, 3289). Thus, we followed these protocols and built a pure

POPC membrane in the simulations. In the revised main text, we have also cited other MD researches without cholesterol in the membrane (page 32).

c. Page 5, lines 149-150: What is the rationale that the authors base their analysis on the RELATIVE movement of TM5 to TM7? To define the outward movement of TM6, the authors refer to a more rigid part of the receptor (i.e. TM2), so why don't they do that for TM5 and TM7 as well? Wouldn't it be wiser to use the same approach to assess the movements of TM5, 6 and 7?

RESPONSE: We have the following reasons to select the activation parameters. First, the motion of TM5 and TM7 alone during activation is not as obvious as the TM6 outward movement, but a relative movement parameter can enlarge this conformational change during activation. Indeed, the range of the distance variation is still smaller (3.9 Å) than the TM6 angle change (32.7°). If the same approach was adopted, the difference would be vague for observation. Second, as the two rigid distances and a TM5-TM7 distance form a specific triangle, it is geometrically the same to use the two distances and a TM5-TM7 distance. Third, the movement of TM5 and TM7 is accompanied by each other, so it is not necessary to measure them distinctly. At last, a free energy landscape should be drawn based on two parameters, but the two distances and an angle have three parameters. Thus, we chose the distance and the angle as the two order parameters to describe the free-energy landscape. Overall, the two order parameters can distinguish the conformational distribution of other class A GPCR structures (Figure 2B).

d. Fig. 2: The projection of previously obtained structures generally confirms the approach of the authors. Though, special attention should be given to the significant outliers in Fig. 2B (e.g. the blue, inactive structure that is almost in the active-state cloud). Are these outliers "exotic" GPCRs that are known to undergo a rather distinct activation mechanism?

RESPONSE: We adopted your suggestion and paid special attention to analyze the significant outliers in Fig. 2B.

Two inactive outliers (human platelet-activating factor receptor (PAFR) with an antagonist SR 27417, PDB ID:5ZKP; P2Y₁₂R with an antagonist AZD1283, PDB ID: 4NTJ) close to the fully active cloud were observed. PAFR undergoes a significant movement of the rigid anchor TM2, so the angle is not in the region of an inactive conformation, but its distance fits the region of an inactive state (19.90, 61.60°). PAFR complexed with an inversed agonist ABT-491 (5ZKQ with a coordinate of (20.30, 44.50°) (Cao et al., *Nat. Struct. Mol. Biol.* **2018**, 25, 488–495) also has a larger angle than other inactive structures. As for the P2Y₁₂R, TM6 moves outwards more than other class A GPCRs in the inactive state, leading to a higher basal activity for activation (K. Zhang et al., *Nature* **2014**, 509, 115–118). Thus, the location of the inactive structure of P2Y₁₂R (21.72, 59.40°) is close to the fully active cloud. Fig. S3 showed structural comparison of the inactive structures of PAFR and P2Y₁₂R with that of AT₁ receptor. In addition, we also analyzed other outliers or the distribution of agonist-bound structures, G protein- or nanobody-bound structures in the free-energy landscape in the revised main text (page 9) and the supporting information (Section 3, SI) as follows:

Section 3: Unusual GPCR Structures on the Free Energy Landscape

Several class A GPCRs have unique activation mechanisms. Thus, they are outliers in Fig. 2B and we show the specific structures in this section. As shown in Fig. S3A, the inactive PAFR (PDB ID: 5ZKP) has a TM2 outward movement, which increases the angle among positions at 6.34, 6.47, and 2.41. For P2Y₁₂R (PDB ID: 4NTJ) in Fig. S3B, the conformation of TM6 is distinctive. N^{6.34} is on the edge of ICL3 and moves outwards, while C^{6.47} becomes close to the center of TM bundles(K. Zhang et al., *Nature* **2014**, 509, 115–118). Thus, the activation angle in both the inactive PAFR and P2Y₁₂R structures is larger than that of other inactive structures.

Fig. S3: The outliers of inactive structures aligned to inactive AT₁ receptor. (A) Intracellular AT₁ receptor (blue) and PAFR (pink) are shown in cartoon. The unique relative position of TM2 is shown. (B) AT₁ receptor (blue) and P2Y₁₂R (salmon) with distinct TM6 twist in its inactive state. Residues for angle measurement on TM6 are shown in sticks.

The active structures defining by receptors with only agonist bound have a broad distribution in Fig. 2B. Half of them are located at the intermediate state, some are close to the inactive cloud, and several of them are in the fully active cloud. This phenomena are attributed to the weak ability of agonists to shift the conformational ensemble and a fully active state needs G protein, β -arrestin, or nanobody to fully stabilize this state (Nygaard et al., *Cell* **2013**, 152, 532–542). Based on the subclass and complex composition, the active structures can occupy different areas. For example, cannabinoid receptors (CBR) structures sometimes have limited TM6 outward movement (Hua et al., *Cell* **2020**, 180, 655–665; Shao et al., *Nat. Chem. Biol.* **2019**, 15, 1199–1205). Thus, 6KQI (CB₁R with an agonist CP55940 and a negative allosteric modulator ORG27569) is located at the region of (19.70, 36.80°) and 6KPC (CB₂R with an agonist AM-841) lies at the region of (19.30, 38.10°). Under different compositions, the more active conformations of the CB₁R structures such as 5XR8

(19.66, 57.74°) and 5XRA (19.72,57.66°) have been solved. Fig. S4 showed structural comparison among different CBRs.

Fig. S4: The outlier example of active structures, CBR. CB₁R-AM-841, CB₂R-AM-841, and CB₁R-CP55940-ORG27569 are depicted as green, brown, and blue cartoons, respectively.

As for the 5-HT_{1B} receptor (PDB ID: 4IAQ) (20.68, 39.26°), it shows obvious extracellular activation features with the shift in the top of TM5, but its intracellular TMs move a little, inferring that the activation signal has not transmitted towards the intracellular side (Wang et al., *Science* **2013**, 340, 610–614). β_1 AR also shows limited TM5 and TM6 movement (4AMI (21.44, 39.98°), 2Y00 (20.11, 41.63°), 2Y01 (20.04, 41.97°), 2Y02 (19.98, 41.28°), 2Y03 (20.02, 41.61°), 2Y04 (20.00, 41.38°)) due to its R-state during activation, reflecting its specific activation mechanism (Warne et al., *Nature* **2011**, 469, 241–244; Warne et al., *Structure* **2012**, 20, 841–849). Other outliers include AT₂ receptor (5UNG (19.17, 61.94°), 5UNH (19.30, 62.21°), 5UNF (19.01,62.78°)), whose TM6 shows an obvious outward displacement when only ligand bound. It has a different activation mechanism and adopts an active-like conformation with agonists (Zhang et al., *Nature* **2017**, 544, 327–332). For the AT₁ receptor's homolog, AT₂ receptor also has a high constitutive activity, related to its

large TM6 outward movement. In addition, NTS₁R is located at the active cloud (constitutively active 5T04 (19.47, 63.27°) and active-like 4GRV (18.64, 66.42°)). Corresponding crystallization articles have stated the high activity (Krumm et al., *Sci. Rep.* **2016**, 6, 38564; White et al., *Nature* **2012**, 490, 508–513). In summary, the positions of active non-rhodopsin structures are related to their conditions and reflect their activity, consistent with the population selection mechanism.

In the fully active structures, rhodopsin needs extra discussion. Fig. S5 shows all active and fully active rhodopsin structures, concentrated at the right of the active cloud. Only 5TE5 (23.31, 40.92°), 6OFJ (20.64, 44.98°), and 5W0P (17.90, 68.30°) are outliers of rhodopsin. The larger distance between TM5 and TM7 reflects the specific activation mechanism of rhodopsin (Zaitseva et al., *JACS* **2010**, 132, 4815–4821).

NTS₁R structures with Gi protein (6OS9 (19.02, 52.31°), 6OSA (19.37, 48.04°) are also situated on the inactive cloud). The smaller TM6 displacement is attributed to the smaller $\alpha 5$ helix of Gi. The non-canonical conformation of 6OSA is close to the inactive conformation with an angle of 48.04°. It is regarded as an intermediate state during activation, but can still bind with a Gi protein (Kato et al., *Nature* **2019**, 572, 80–85).

Fig. S5: Projection of all reported fully active (red) and active (green) structures of rhodopsin onto the AT₁ receptor conformational landscape. The unit of free-energy values is kcal/mol.

e. The key reference 21 does not report any MD.

RESPONSE: We have corrected it in the revised main text.

f. The authors introduce the topic of receptor activation arguing that it is ligand-driven (see my comment above). The inactive structure that is the starting point of the NEB and MD is antagonist bound, whereas the defined endpoint refers to an agonist- and nanobody bound active structure. However, the authors do not mention if either the antagonist has been removed for the MD or if the agonist was introduced in the inactive structure to facilitate the activation process. If ligands are not present, the authors observe constitutive activity in their MD; this must be clarified. The authors should provide experimental (wet lab) data providing the notion of AT₁R constitutive activity. This is not only to confirm their findings from the in-silico experiments, but it would also indicate which signaling pathways this active receptor state reflects (AT₁R can couple to multiple G protein isoforms and b-arrestin). At least a paper should be cited, where constitutive activity of AT₁ receptor was investigated.

RESPONSE: Thanks for your suggestion. Considering our *apo* structure in MD simulations, AT₁ receptor is anticipated to have basal activity. The constitutive activity of AT₁ receptor has been proved by the BRET assay, in consistent with previous researches (Karnik & Unal, *Hypertension* **2012**, 59, 542–544; Unal & Karnik, *Adv. Pharmacol.* **2014**, 70, 155–174).

We have stated that antagonist, agonist, and nanobody were all removed before NEB and MD simulations, namely the *apo* structures were used in our *in silico* research (see Results, page 5 and Methods, page 30).

AT₁ receptor can be activated in the absence of AngII (Karnik & Unal, *Hypertension* **2012**, 59, 542–544; Unal & Karnik, *Adv. Pharmacol.* **2014**, 70, 155–174). We have performed the BRET experiment to confirm the constitutive

activity of the AT₁ receptor in Gq, Gi, and G12 compared with a control vasopressin 2 receptor (V₂R) (Fig. 3C-3E). This result has been added in the revised main text as follows (pages 11 and 12):

As our simulations were based on the *apo* AT₁ receptor, the spontaneous activation reflects a constitutive activity of the wild-type (WT) AT₁ receptor, which has been previously reported (Karnik & Unal, *Hypertension* **2012**, 59, 542–544; Unal & Karnik, *Adv. Pharmacol.* **2014**, 70, 155–174). Using bioluminescence resonance energy transfer (BRET) assays in G proteins, we confirmed the constitutive activity of the AT₁ receptor in Gq, Gi, and G12 compared with a control vasopressin 2 receptor (V₂R). We first controlled the equal levels of the AT₁ receptor and V₂R on the cell surface (Section 6, SI). Next, the dissociation of G α and G $\beta\gamma$ was represented by the decrease of Δ BRET. As shown in Fig. 3C–E, Δ BRET decreased with increasing AT₁ receptor levels in a dose-dependent manner, suggesting that downstream G proteins are dissociated from the *apo* AT₁ receptor. In contrast, V₂R levels had no influence on the Δ BRET value. Collectively, these data suggested the constitutive activity of the AT₁ receptor.

Fig. 3: The three macrostates divided by Markov state model (MSM) and constitutive activity. (A) The distribution of three macrostates on the free energy landscape. The attribution and probability of each macrostate is shown on the right. (B) The transition time among the active, inactive, and intermediate states, represented by the mean first passage time. (C-E) Constitutive activities of AT₁ receptor in Gq (C), Gi (D), and G12 (E) pathways. Vasopressin 2 receptor (V₂R) was used as a negative control. Data were from three independent experiments. The bars indicate the mean \pm SEM values.

g. Supplementary table Section 2 S1: I just quickly looked at this eminent summary of GPCR structures and was surprised to find only one AT1 receptor structure. Lacking in the table: 4YAY, 5UNG, 6OS0, 6OS1, 6OS2. Are those structures included in the data in Fig. 2B? Did not check the rest of the table.

RESPONSE: We have checked Table S1 again and added the corresponding structures. The 6OS series are not released when we collected the data in Table S1. Now, we have updated Table S1 and Figure 2B with all GPCR structures released until April 26, 2021 and included all PDBs mentioned here.

h. In Section 2. What is meant by “the state of the protein referred to the gpcrdb database”? No column in the Table S1 has the title “state of protein” – do the authors refer to activity? Also, given the rapid pace of the appearance the authors should state a date of the table, especially since they claim comprehensiveness.

RESPONSE: The state of protein in the original manuscript refers to activity. However, according to (Latorraca et al., *Chem. Rev.* **2016**, 117, 139–155; Manglik et al., *Cell* **2015**, 161, 1101–1111; Nygaard et al., *Cell* **2013**, 152, 532–542), we have changed our definition of activity and revised the so-called gpcrdb definition. In the new definition of activity, we defined the GPCR structures with inverse agonist- or antagonist-bound structures as inactive conformations, with only agonist-bound structures as active conformations, and with both agonist- and G protein-, β -arrestin, or nanobody-bound structures as fully active conformations. This definition is more

reasonable. Furthermore, as shown in Section 2 (SI), the date (April 26, 2021) of Table S1 has been provided.

i. In the context of defining active state, the authors need to provide the information if “active” means agonist and G protein (mG, nanobody)-bound or only agonist-bound receptor in Table S1. As mentioned above, an agonist-bound structure is only fully active in the presence of the G protein (or mG, arrestin, nanobody)

RESPONSE: Thanks for your kind remind. In our new definition of activity, we defined the GPCR structures with inverse agonist- or antagonist-bound structures as inactive conformations, with only agonist-bound forms as active conformations, and with both agonist- and G protein-, β -arrestin, or nanobody-bound forms as fully active conformations. We have revised this definition in the main text (pages 5 and 8) and the supporting information (Section 2, Table S1).

j. Out of curiosity: are the arrestin structures clustered and are they clustered at a different region compared to G protein-bound structures? Where are nanobody-bound structures (do nanobody structures sample both arrestin and G protein cluster?).

RESPONSE: Non-rhodopsin receptor structures with G protein bound are mostly distributed in the active cloud, while the structures of rhodopsin-G protein complex are situated on the right of the active basin (Fig. S6A). Gs protein has a bulkier $\alpha 5$ helix than Gi/o protein, which causes a larger TM6 movement of the receptors upon Gs binding than Gi/o. Thus, GPCR-Gs structures mostly cluster at the top right of the active cloud and GPCR-Gi/o structures lie in a lower position with a smaller angle. As for the TM5-TM7 movement, the distribution of Gi/o bound structures is dispersed compared with the Gs bound structures, indicative of less restraint of TM5-TM7 movement in response to Gi/o binding.

Currently, only four non-rhodopsin structures with β -arrestin bound (6U1N, 6PWC, 6UP7, and 6TKO) have been solved (Fig. S6B), so the distribution tendency is not clear. However, it is inferred that β -arrestin binding leads to a smaller TM6

displacement with Gi/o binding rather than Gs binding (García-Nafriá, et al., *Nature*, **2018**, 558:620-623).

As for the nanobody-bound receptor complexes (Fig. S6C), some of them cluster in the position close to GPCR-Gs complexes because they are Gs-mimic nanobodies. Nanobody-bound receptor structures can sample both the β -arrestin- and G protein-bound structure clusters. Remarkably, some nanobodies can stabilize the intermediate (nanobody 6 for succinate receptor SUCNR1) or the inactive (nanobody 6 for κ -OR and nanobody 60 for β 2AR) conformations. These receptor structures are located at a common position for corresponding structures without nanobody.

Finally, we have added the discussion of the distribution of receptor-G protein and receptor-nanobody complexes in the free-energy landscape in the revised main text (page 9) and supporting information (section 4, SI).

Fig. S6: Projection of all reported G protein- (A), β -arrestin- (B) and nanobody- (C) bound GPCR structures onto the AT₁ receptor conformational landscape.

k. Page 6 – second paragraph, what about fully inactive, inverse agonist-bound structures?

RESPONSE: In the new definition, we defined inverse agonist-bound structures as inactive conformations. We projected these structures on the free-energy landscape (Fig. R1). Inverse agonist-bound non-rhodopsin GPCRs are located at the left of the inactive cloud, while inverse agonist-bound rhodopsin GPCRs are situated on the right of the inactive cloud due to its longer TM5-TM7 distance. Several structures

show a highly small TM6 angle compared with others, indicating their obvious inactivity.

Fig. R1: Projection of all reported inverse agonist-bound GPCR structures onto the AT₁ receptor conformational landscape.

1. On page 8 the authors finally mention that the analysis was done in the apo state. Thus, the authors observe constitutive rather than agonist-induced activation. This must be stated clearly. On the other hand, this might offer space for control experiments. Could the authors employ an intermediate state and add agonist/inv agonist and follow activation towards more active and inactivation towards inactive as defined in Fig. 3?

RESPONSE: In our revised manuscript, the constitutive activity has been clearly stated and proved. We have also supplied additional *in silico* experiments validating that the intermediate state is activated with an agonist bound and deactivated with an inversed agonist bound.

We have used BRET to validate the constitutive activity of the AT₁ receptor in the revised main text (pages 11 and 12) and have answered this question in previous **comment f**.

We have also conducted additional MD simulations to validate the intermediate state. We docked the inverse agonist olmesartan and the natural agonist AngII to the orthosteric pocket of the intermediate AT₁ receptor, respectively. After the same

simulated protocol, we performed 2 μs conventional MD simulations followed by 1 μs gaussian accelerated MD (GaMD) simulations for three independent rounds with random velocities to accelerate simulation sampling. After simulations, we calculated the activation parameters and plotted the free energy landscape of each round in both AngII- and olmesartan-bound AT₁ receptor (Fig. S13).

In the AngII binding, the TM6 angle of the AT₁ receptor largely adopted more than 55° and the tendency fitted the outward movement of TM6, which resembled to the active state. On the contrary, olmesartan binding rendered the AT₁ receptor less active in the TM6 angle and moved the free-energy landscape downwards. As for the distance index, major conformers (darkest part) of the AngII-bound AT₁ receptor were around 15-17 Å, which are close to the active conformation. However, olmesartan binding maintained the distance value in the range of 18-20 Å, consistent with the feature of the inactive conformations. Thus, inverse agonist and agonist binding shifted the intermediate state towards the inactive and active states, respectively. Together, these results suggested the reasonability of the intermediate structure captured from the transition pathway.

Finally, we have added the above result in the main text (page 15) and in the supporting information (Section 11, SI).

Fig. S13: The conformational landscape of the AT₁ receptor generated using the C α atom distance between L^{5.55} and N^{7.46}, and the angle among the C α atoms of F^{6.34}, S^{6.47}, and V^{2.41} as the order parameters along the activation pathway. (A-C) GaMD rounds 1-3 in the intermediate AT₁ receptor with the agonist AngII bound. (D-F) GaMD rounds 1-3 in the intermediate AT₁ receptor with the reverse agonist olmesartan bound.

m. Page 9, lanes 261 downward: Is the ECL2 closure of the orthosteric site driven by the G protein mimicking nanobody as reported in the allosteric coupling paper from DeVree et al., Nature, 2016? Why does the upward movement of Helix 8 open a ligand binding pocket? The authors should provide a better rationale (maybe with additional figures) why they are concluding this here. And most importantly: AT₁ receptors can activate multiple G protein subtype upon agonist binding (G12/13, Gi/o, Gq). Is the transition of AT₁R to the active state through this proposed intermediate state required for the activation process of all signaling pathways? A combination of in-silico and in-vitro experiments is needed to confirm this core statement of the manuscript. (Suggestion for in-vitro experiments: G protein sensor activation upon stimulation of AT₁R mutants where the key microswitches required for the intermediate state are abolished).

RESPONSE: Thanks for your great suggestions and we respond to these comments one by one. As for the most important issue, we designed mutants for intermediate-specific microswitches and both the constitutive and AngII-induced G protein activities of the variants decreased. Thus, the intermediate state is required for all G protein pathways.

1. ECL2 movement

According to the population shift mechanism, the binding of a nanobody changes the conformational distribution of GPCRs towards the fully active state. Of note, the active state naturally exists in the ensemble but occupies a limited population (Latorraca et al., *Chem. Rev.* **2016**, 117, 139–155; Nygaard et al., *Cell* **2013**, 152,

532–542). Thus, the orthosteric pocket-closed conformation of the receptors induced by the nanobody binding can be observed in the simulations. We have shown a zoom-in view in Fig. S9A to describe the movement of TM6 and ECL2, which closed the orthosteric pocket. In the revised main text, we have stated that the orthosteric pocket closure property has also been observed in the active β_2 AR-nanobody complex structure (DeVree et al., *Nature* **2016**, 535, 182–186) (page 14).

2. H8 movement

We suggested the possibility of cryptic pocket formation due to the movement of H8, which was confirmed by pocket prediction and experiments. We have provided a supporting figure to describe the movement of H8 during activation. As shown in Fig. S9B, H8 forms a large angle with TM7, which is unable to accommodate a ligand in the inactive structure. Moreover, the active macrostate has a tight space between TM1, TM7, and H8 that a pocket cannot be formed. Thus, we inferred that a pocket may exist among TM1, TM7, and H8 in the intermediate state. Finally, we have revised the sentence in the main text (page 14) and the supporting information (section 7, SI) as follows:

In addition, H8 moved upward from the inactive to the active state to provide space for the downstream effectors. This upward movement produces a suitable volume for ligand binding among TM1, TM7, and H8, which may offer an opportunity for the formation of a “cryptic” pocket for drug design (Section 7, SI).

Fig. S9: (A) Extracellular view of the macrostates. Arrows show movement during activation. N-terminal, TM1, TM2, and ECL1 are hidden for clarity. (B) The position of H8 and surrounding TMs in the macrostates.

3. AT₁ receptor activates multiple G protein subtypes upon agonist binding and the key micro-switches was identified

We compared the three macrostates and found several intermediate-specific micro-switches (Fig. S14). As shown in Section 12 (SI), these micro-switches exist in the intermediate state due to their unique conformation. During activation, TM5 moves towards TM7 and becomes close to TM6 in its intermediate state, in which the polar contacts among K^{5.42}, H^{6.51}, and T^{6.55} form (Fig. S14A). In addition, the TM6 outward movement generates a hydrogen bond between I^{6.37} and Y^{5.58} in the intermediate state (Fig. S14B). TM6 movement also causes the formation of a hydrophobic network among M^{6.38}, W^{5.62}, and F^{6.34} in the intracellular intermediate structure (Fig. S14C). As for the H8 movement, the hydrophobic contacts among V^{1.53}, V^{1.56}, and F^{8.50} only exist in the intermediate state (Fig. S14D).

Based on these interactions, we designed single mutations K^{5.42}A, Y^{5.58}A, W^{5.62}A, F^{6.34}A, and F^{8.50}A. After controlling the equal expression levels between the variants and the WT receptor, we measured both constitutive and AngII-induced G protein

activities using BRET assays. The result of BRET assays was shown in Fig. 5 and the summarized data were listed in Table S2.

Based on BRET assays, the disruption of micro-switch interactions in the intermediate state inhibited both the constitutive and AngII-induced activities of the AT₁ receptor for G proteins, including Gq, Gi, and G12 (Fig. 5). The decrease of constitutive activity was more obvious than the AngII-induced activity. In the AngII-induced activity, K^{5.42}A decreased the activity of Gq and Gi more than G12, suggesting that Gq and Gi activity may need a larger movement of extracellular TM5. Although W^{5.62}A and F^{6.34}A did not influence the Gq and Gi signal in E_{max}, the increase of the EC₅₀ value reflected a weaker activation upon AngII binding (Section 14, SI). The remaining Y^{5.58}A and F^{8.50}A mutations similarly modulated the activity of the AT₁ receptor in the three G proteins. In summary, these results indicated that the intermediate state is required for G protein signaling pathways, highlighting its possibility to become a drug target.

Finally, we have added the above results in the revised main text (pages 15-17) and the supporting information (Sections 13 and 14, SI).

Fig. S14: Intermediate-specific interactions (cyan) in other two active (orange) and inactive (blue) states. (A) The polar contacts among $K^{5.42}$, $H^{6.51}$, and $T^{6.55}$. (B) The hydrogen bond between $Y^{5.58}$ and $I^{6.37}$. (C) The hydrophobic network among $M^{6.38}$, $W^{5.62}$, and $F^{6.34}$. (D) The hydrophobic contacts among $V^{1.53}$, $V^{1.56}$, and $F^{8.50}$.

Fig. 5: (A) The intermediate-specific micro-switches. Involved residues are shown in sticks and corresponding distances are depicted by yellow dashed lines. (B-D) Constitutive and AngII-induced activities of WT AT₁ receptor and mutants in Gq (B), Gi (C), and G12 (D) pathways. Data were from three independent experiments and representative dose-response curves were shown in AngII-induced activity.

Table S2. The EC₅₀ and E_{max} values of the BRET assay for Gq, Gi, and G12.

	Gq		Gi		G12	
Mutation	EC ₅₀ (nM)	E _{max} (%)	EC ₅₀ (nM)	E _{max} (%)	EC ₅₀ (nM)	E _{max} (%)
WT	8.87±0.49	100	10.09±0.75	100	11.32±0.69	100
K ^{5.42} A	20.92±1.40**	40.33±2.67**	222.70±11.42***	43.33±1.20***	14.18±0.61*	70.00±1.00**
Y ^{5.58} A	50.45±0.50***	53.33±0.33***	105.00±0.75***	54.67±1.33***	17.07±0.67**	29.67±0.33***
W ^{5.62} A	19.87±0.71***	96.00±1.00	16.77±0.14***	116.00±3.00	24.39±0.54***	33.67±0.33***
F ^{6.34} A	32.92±1.30***	88.00±5.51	21.18±0.90***	104.30±2.72	14.79±0.67*	46.67±0.33***
F ^{8.50} A	16.08±0.44***	85.00±1.16**	20.18±0.50***	82.00±2.65*	156.50±5.40***	33.67±0.33***

n. In the large table S1 it is instrumental (as mentioned above) to be absolutely clear about what is meant with “active”, “inactive” and “intermediate” receptor. For that purpose, the authors must list presence of e.g. G protein, nanobody, mini G protein, arrestin etc. The referral to the gpcrib interpretation of activity is not sufficient. I only went in to some examples and found e.g. EP3 (PDB6AK3), a receptor that is listed with active activity in the absence of G protein or nanobody, whereas a β_2 AR (4LDO) is equally listed as active but has a nanobody bound. I fear that this poorly defined handling of receptor activity compromises the analysis and especially the MSM presented in Fig. 3. How does the MSM handle the difference in active state given presence and absence of G protein? The authors discuss the relevance of constitutive activity shortly on page 9, but in the remaining text, the differentiation between basal activity vs agonist-induced and G protein-stabilized activity remain obscure.

RESPONSE: Thanks for your carefulness and kind suggestion. We have carefully revised the Table S1 and defined the activity of receptors as below:

We defined inverse agonist- or antagonist-bound forms as inactive conformations, only agonist-bound forms as active conformations, and both agonist- and G protein-, β -arrestin, or nanobody-bound forms as fully active conformations.

We have also listed the existence of substrates or antibodies (including nanobodies) in Table S1 for the fully active structures. In our new definition, EP₃R (6AK3) is regarded as an active conformation because it is agonist-bound but not substrate-stabilized. β_2 AR (4LDO) is a fully active conformation because it binds with both an agonist epinephrine and a nanobody 80. Other previous vague proteins have also been renewed in the Table S1.

As for the MSM analysis, it does not deal with crystal structures but analyzes MD trajectories. The input of MSM is the activation parameters from 300 μ s MD trajectories, while the output is the dynamic parameters for transfer among the three states. Thus, MSM does not need to deal with structural information in the Table S1.

We have validated the basal activity of the AT₁ receptor using BRET assays (pages 11 and 12). Comparing our active cloud and the fully active structures, we suggested that transducers are necessary to stabilize the fully active conformations of GPCRs, which has been added in the revised main text as follows (page 9):

Since our simulations are based on the *apo* structure, the active cloud is not highly overlapped with fully active non-rhodopsin structures, indicative of the instability of active structures and the necessity for transducers to stabilize the fully active conformations.

For the differentiation between the basal activity vs the agonist-induced activity, we have answered this question in **comment m**. We performed BRET assays to compare the constitutive and AngII-induced activities of the AT₁ receptor for G proteins, including Gq, Gi, and G12 (Fig. 5). The decrease of constitutive activity was more obvious than the AngII-induced activity. We have added the above results in the revised main text (pages 15-17) and the supporting information (Sections 13 and 14, SI).

o. Page 11: Referral to ref 27 and the “canonical active” conformation adds to the confusion of what the authors define as “active”. (btw the authors should consider the use of the term “canonical” – in science history it has happens that findings that appear canonical one day aren’t anymore the next day). Maybe that should be replaced by G protein-bound or something in that direction.

RESPONSE: Thanks for your comments. We have changed the “canonical active” conformation by the Gq-bound conformation in the revised main text.

p. Page 12: Modelling of active conformations with G protein and arrestin. These in silico experiments need to be explained better. Since the macrostates were developed in the absence of G protein/arrestin the modeling of the complex appears to this reviewer as a weak argument for bias. Experimental evidence from in-vitro studies is

needed to support this finding. NMR studies with purified and labeled AT1R would probably allow for the most direct validation of these proposed macrostates but also signaling (=activation) readouts with AT1R mutants could further underline the authors' claims. The authors neglect the fact that AT1R can also couple to G proteins from the G12/13 and Gi/o family. Given that they claim at least 8 macrostates of this receptor (four of them being at least partially active), it is essential to also assess (computationally and experimentally) their effects on other signaling pathways than Gq and beta-arrestin.

RESPONSE: Thanks for your good suggestion. We have conducted the BRET assay using the AT₁ receptor with macrostate-specific microswitch mutations to provide clues for the biased signaling of the AT₁ receptor. BRET assay inferred that macrostates 6 and 7 are biased toward Gq and β-arrestin 2 pathways, consistent with our structure model. As for macrostates 1 and 5, they are related to Gi and G12 signal, respectively. Because biased signaling is not the main part of our research and our lab does not involve in NMR experiments, we did not conduct NMR. Alternatively, we designed AT₁ mutants and performed the BRET assay using the AT₁ receptor with macrostate-specific micro-switch mutations to provide clues for the biased signaling of the AT₁ receptor (pages 18-21 and Section 17, SI, shown as follows). In the Discussion part, we stated that we just provided indication for the biased signaling of the AT₁ receptor, which should be further confirmed by NMR or other experiments (page 28).

Since AT₁ receptor has a constitutive activity for G proteins (Fig. 3C-3E) and showed β-arrestin activity upon AngII binding or specific agonists (Ryba et al., *Circulation* **2017**, 135, 1056–1070; Suomivuori et al., *Science*. **2020**, 367, 881–887; Wingler et al., *Science* **2020**, 367, 888–892), we explored the connection between different active states and biased signaling. Because Gq is the major G protein activated by AT₁ receptor and β-arrestin 2 is commonly used in biased signaling (Griendling et al., *Hypertension* **1997**, 29, 366–373; Ohtsu et al., *Endocrinology* **2008**, 149, 3569–3575; Suomivuori et al., *Science*. **2020**, 367, 881–887; Zhang & Shi, *J. Immunol. Res.* **2016**, 3969023), we first constructed AT₁ receptor–Gq and AT₁

receptor- β -arrestin 2 complexes based on the active macrostates 6 and 7 to determine whether the two active conformations had a bias for the transducers (Section 16, SI). The models showed that macrostates 6 and 7 tend to initiate Gq and β -arrestin 2, respectively, suggesting that biased conformations naturally exist in the ensemble of the AT₁ receptor.

Supported by this preliminary analysis, we further investigated biased signaling in the AT₁ receptor based on tICA analysis. From the representative structures, we explored specific micro-switches of each macrostate (Fig. 7A). Y^{7.53} is close to hydrophobic residues at TM1 and TM2 in both macrostates 5 and 6, while it forms a hydrogen bond with R^{3.50} or Y^{5.58} in macrostates 1 and 7, respectively. Thus, by introducing a hydrophobic residue, the Y^{7.53}I mutation could maintain the hydrophobic interactions, but disrupted the polar interactions, leading to the stability of macrostates 5 and 6, and the instability of macrostates 1 and 7. In addition, N^{3.35} and D^{2.50} form a tight hydrogen bond in the macrostates 7, whereas it is weak in macrostate 6 and diminishes in macrostates 1 and 5. Introduction of N^{3.35}A mutation disrupted the hydrogen bonding interaction, preferred the conformation with a long distance between the residue at 3.35 and the polar residue at 2.50 (macrostates 1, 5, and 6). In turn, a moderate mutation D^{2.50}N changes the charge of its sidechain, which may weaken the interaction between residue at 3.35 and residue at 2.50. Because the macrostate 7 has a strong hydrogen bond, its conformation may maintain in response to the D^{2.50}N mutation. However, macrostate 6 may be disturbed owing to this mutation. Our mutation experiments for biased signaling were benchmarked against these micro-switches and the corresponding results were shown in Fig. 7B–E. An operational model was also applied to determine the biased signaling with normalized E_{max} value (Fig. 7F).

Fig. 7: (A) Macrostates 1 (green), 5 (salmon), 6 (gray), and 7 (purple) with their corresponding specific micro-switches are shown in cartoon. Involved residues are shown in sticks. Because the fluctuation of the distance between N^{3.35} and D^{2.50} differs between structures, the distance is shown in mean ± standard deviation form for all snapshots from the representative trajectories of macrostates. (B-E) AngII-induced Gq activation (B), β-arrestin 2 recruitment (C), Gi activation (D), and G12 activation (E) in HEK293 cells transfected with WT AT₁ receptor or mutants. Three independent experiments were performed and representative dose-response curves were shown. (F)

Heat map of AT₁ receptor signaling signatures of WT AT₁ receptor and mutants. The relative activities $\Delta\log(\tau/K_A)$ of WT AT₁ receptor and mutants in each signaling pathway calculated in (Section 17, SI) were expressed as a heat map.

In Fig. 7F, blue and light pink represent a weak signal and red reflects a strong signal, compared with the WT AT₁ receptor. In particular, N^{3.35}A mutation (benefits macrostates 1, 5, and 6, inhibits macrostate 7) obviously promoted Gq and Gi signals but inhibited G12 and β -arrestin 2 signals. In contrast, D^{2.50}N showed relatively weaker Gq and Gi signals compared with G12 and β -arrestin 2 signals (Section 17, SI). Facilitating macrostates 5 and 6, Y^{7.53}I also promoted Gq but inhibited β -arrestin 2. Thus, micro-switches facilitating macrostate 6 and restraining macrostate 7 (N^{3.35}A, Y^{7.53}I) led to Gq activation and β -arrestin 2 inhibition, whereas the D^{2.50}N mutation, boosting macrostate 7 and reducing macrostate 6, preferred β -arrestin 2 rather than Gq signal. This is in line with our hypothesis that macrostate 6 is biased to the Gq pathway, whereas macrostate 7 is for β -arrestin 2 (Section 16, SI). As for Gi, it can be inferred that the intermediate macrostate 1 (inhibited by Y^{7.53}I and D^{2.50}N, facilitated by N^{3.35}A) may play a critical role in the activation toward Gi since Y^{7.53}I and D^{2.50}N suppressed Gi signal but N^{3.35}A increased it. Conversely, G12 activation may be related to the macrostate 5 (facilitated by Y^{7.53}I, inhibited by N^{3.35}A and D^{2.50}N) because Y^{7.53}I stimulated the G12 signal, while N^{3.35}A and D^{2.50}N repressed G12 activation.

In summary, based on the specific micro-switches of each macrostate, we designed mutants and tested downstream activity to measure the biased signaling in each state. Macrostates 6 and 7 were responsible for the biased signaling of Gq and β -arrestin 2, respectively. In addition, it was inferred that Gi and G12 signals are related to the existence of macrostates 1 and 5, respectively.

Finally, we have added the above results in the revised main text (pages 18-21) and the supporting information (Section 17, SI).

q. Discovery of a cryptic allosteric site (please add references to the approach Fpocket). Four of the identified pockets (P2, P3, P4 and P5) are localized on the outer surface of the receptor. How can this be defined a binding "pocket"? The authors should elaborate more on the methodological details of their Fpocket approach. The statement that the authors discovered a new, likely allosteric site appears weak, without an actual proof to target this site with a small molecule. At least a docking analysis could be provided (even though I understand that this is out of the scope of the paper). It appears rather like a well-founded hypothesis that requires testing. In Fig 9A, the authors could highlight the residues that are creating the hypothesized P6 allosteric site for clarity. The mutational alanine mutation appears like a very blunt tool to argue for the existence of a cryptic allosteric site in a mutational cluster. Are all these 5 clusters really involved in the formation of P6? From the structures in Fig. 8 and Fig. 9, I would think that it is only one TM (TM1 or TM2?) + TM7 + H8, meaning only three clusters should be examined. The authors do not explain why they chose these 5 clusters and therefore the correlation between their signaling data and P6 formation is substantially hampered! While the authors claim that cluster mutation underline the hypothesis of an allosteric P6 ligand binding pocket, I wonder whether this is this because a) P6 is not a "pocket" but rather a microswitch or b) because an endogenous allosteric modulator binds to P6 in wildtype but not mutant AT1R?

RESPONSE: Thanks for your good suggestion. We have briefly described the algorithm of the extensively-used pocket discover program, Fpocket, and conducted a glide docking to prove the rationality of P6 by reasonable binding pose. We also explained why the five clusters were previously designed and proved the existence of P6 by seven new single-point mutations.

Fpocket is a commonly-used pocket discovery approach in the structure-based drug design (Qiu et al., *J. Med. Chem.* **2020**, 63, 3665–3677; Ni, et al. *Chem. Sci.* **2021**, 12, 464–476). It defines a sphere that contacts four atoms on its boundary and contains no internal atom as an alpha sphere. Thus, clefts and cavities tend to have alpha spheres smaller than the exterior and larger than the interior of the protein. Also, clefts have more alpha spheres than other places. It estimates the position of alpha

spheres by Voronoi tessellation. Fpocket then selects alpha spheres defined by zones of tight atom packing. In the following cluster step, it excludes large spheres at the protein surface which solely composes a sphere cluster, then aggregates clusters with a close center of mass to a large cluster. Finally, a multiple linkage clustering approach is used to further merge clusters. This process is applied to remove hydrophilic or small putative binding pockets. At last, the ability to bind a small molecule was evaluated by Partial Least Squares fitting to pocket descriptors, and top-scored pockets are shown. We have cited the corresponding article and briefly described the Fpocket algorithm in the revised main text (page 22).

We performed additional molecular docking aiming at the cryptic allosteric site using the allosteric GPCR sub-library of Enamine and identified some compounds. In Section 19 (SI), the best-scored compound Z367028310 was identified and its binding mode was showed. This compound has many hydrophobic contacts with surrounding residues of the AT₁ receptor and the binding pose highly overlaps with P6. We have added this result in the revised main text (page 24) and the supporting information (Section 19, SI).

As for the five mutation clusters, we are sorry for our previous vague explanation. Three of them are exactly on P6 and the other two clusters are located at the pathway from P6 to the G protein pocket. The design of P6 site-direct mutations (clusters 1, 4, and 5) is easy to be understood, as the direct perturbation at P6 influences the G protein pocket. Blocking the pathway from P6 to G protein pocket by our mutations (cluster 2 and 3) is anticipated to prove that the signal pathway exists between the two pockets. However, this indirect confirmation is kind of weak, so we deleted this part in the revised manuscript and focused on the direct influence of P6 on G protein binding.

We designed point mutations with the help of the AlloSigMA server and the result of our molecular docking. AlloSigMA predicts the allosteric perturbation between areas (Guarnera et al., *Bioinformatics* **2017**, 33, 3996–3998; Tan et al.,

Nucleic Acids Res. **2020**, 48, 116–124). AlloSigMA analysis indicated that ligand binding on P6 influenced the G protein pocket of the receptor and that stable/bulky or unstable/tiny mutations on several P6 residues changed the dynamics of the G protein pocket (Section 21, SI). With the help of guidance, we designed bulky mutations (G^{1.49}L, F^{7.55}W, and F^{8.54}W, Fig. 10A) and tiny mutations (F^{1.48}A, N^{7.49}A, Y^{7.53}A, and F^{8.50}A, Fig. 10B) in the AT₁ receptor and used BRET assays to test the regulation ability of P6 on both G protein and β -arrestin pathways. Under similar protein levels (Section 22, SI), point mutations disturbed both Gq and β -arrestin 2 activities (Fig. 10C, 10D, and Section 22, SI). Compared with the Gq signal, β -arrestin 2 signal highly decreased in the presence of the mutations, indicating that P6 is a potential G protein-biased allosteric site. In all Gq mutations, although the F^{8.54}W mutant retained the same E_{max} as the WT AT₁ receptor, the higher EC₅₀ value suggested a weaker effect of AngII. Thus, point mutations guided by AlloSigMA suggested the existence of P6. Related results have been added in the revised main text (page 26) and in the supporting information (Sections 21 and 22, SI).

Fig. 10: (A-B) The position of bulky (A) and tiny (B) mutations around P6. (C-D) AngII-induced Gq activation (C) and β-arrestin 2 recruitment (D) in HEK293 cells transfected with WT AT₁ receptor or mutants. (E) AngII-induced conformational change of AT₁ receptor Gq sensor in HEK293 cells transfected with AT₁ receptor Gq sensor or sensor-based mutants. Three independent experiments were performed, and representative dose-response curves were shown in (C-E). The data summary was shown in Section 22, SI.

r. Regarding the functional readout of G protein activation and arrestin recruitment. The G protein readout is based on a decrease in BRET as a result of Gq dissociation (TRUPATH sensors). The arrestin recruitment is based on a direct BRET between YFP-tagged receptors and Rluc-tagged arrestin. While the authors argue about receptor conformations in this manuscript, none of these assays probes for receptor conformation (the arrestin could be a matter of discussion – I agree). I would suggest to rather use BRET probes that act as sensors of receptor conformation, such as a tagged nanobody or a tagged mini G protein. In the figure legend, no information about experimental repetitions, curve fitting, normalization is provided.

RESPONSE: We appreciate your suggestion and have conducted intramolecular FIAsh-BRET assay using our recently developed AT₁ receptor conformational sensor (Fu et al., *Cell Res.* **2021**, doi:10.1038/s41422-020-00464-8; Li et al., *Nat. Commun.* **2018**, 9, 11). As expected, AngII stimulation on the AT₁ receptor Gq sensor decreased the BRET signal in a dose-dependent manner, suggesting an increase of the distance between the C-terminus and the FIAsh motif inserting at ICL3 of the AT₁ receptor (Figure 10E, 10F). Notably, mutations at all the seven sites led to the impairment of AngII-induced AT₁ receptor conformational change as revealed by the increased EC₅₀, while the five mutants, including G^{1.49}L, F^{1.48}A, N^{7.49}A, Y^{7.53}A, and F^{8.50}A, also displayed decreased maximal response (Section 22, SI). These data suggested that the potential allosteric site participated in AT₁ receptor signal regulation through conformational modulation. Thus, we confirmed that P6 influences the dynamics of the G protein pocket.

In addition, we have also provided the repetitions, curve fitting, and normalization information in our figure legends.

Related results have been added in the revised main text (pages 25-27) and in the supporting information (Section 22, SI).

Minor comments:

1. The official nomenclature of the angiotensin II type 1 receptor according to IUPHAR is AT₁ receptor (number in subscript) – this should be used throughout the work.

RESPONSE: We have changed all AT₁R to AT₁ receptor in revised manuscript. We have also revised all GPCR names in the main text and Table S1 to meet the official name according to IUPHAR nomenclature.

2. 1st sentence of the abstract: receptors are described as frequent targets – this feels awkward, rather “are the most frequently targeted...”

RESPONSE: This sentence has been replaced by “G protein-coupled receptors (GPCRs) are the most frequent proteins targeted by approved drugs”.

3. G protein rather than G-protein

RESPONSE: All “G-protein” has been replaced by “G protein”.

Reviewer #2 (Remarks to the Author):

In the manuscript titled “Activation pathway of a G protein-coupled receptor uncovers conformational intermediates as novel targets for allosteric drug design”, the authors used AT1R as the prototype GPCR, and used a computational framework including transition pathway generation program NEB and simulated annealing to generate 15 structures on the MEP, and followed by 10 2 μ s- unbiased MD runs for each structures (totally 300 μ s) to investigate the landscape of its dynamic activation pathway. For the MSM analysis of the comprehensive simulations, they discovered canonical (Gq/arrestin balanced) and alternative (arrestin biased) active states, which is in consistent with the recently reported biased agonism study of AT1R by Ron Dror et al (Science, 2020, 367, 881-887). In conformational intermediates, the authors identified several allosteric pockets and reveals P6 as the novel cryptic allosteric binding site for AT1R (although it exists in several other GPCRs’ intermediates) that could be used to develop allosteric modulators of AT1R. Furthermore, they validated the P6 site through clustered alanine scanning and BRET.

In summary, this is a compelling study of AT1R’s landscape of dynamic activation pathway. I have the following minor concerns:

RESPONSE: We heartfully appreciate the time that the reviewer has dedicated to providing insightful suggestions on ways to improve our manuscript. According to the reviewer’s comments, we have carefully revised the manuscript and hope our detailed responses address your comments.

1. In line 148 and 150, the N7.46 seems should be N7.49.

RESPONSE: We have checked the position of activation parameters and confirmed that it is N^{7.46}. We chose this residue because N^{7.46} is located at the TM7 twist position, which reflects the inward movement of TM7 during activation during MD simulations. We also state the reason why we chose N^{7.46} in the revised manuscript (page 6).

2. In Figure 4D, the residue L6.30 should be N6.30.

RESPONSE: In Figure 4D and footnotes, L^{6.30} has been corrected to N^{6.30}.

3. In line 247, Y5.53 should be Y5.58.

RESPONSE: Y^{5.53} has been corrected to Y^{5.58}.

4. In line 245, please rationalize the chosen of CD atom of R3.50. Why not CZ of R3.50?

RESPONSE: Thanks for your kind suggestion. In line 245, we mentioned that the distance between R^{3.50} and N^{6.30} was measured, and the specific atoms were depicted in Figure 4 legend. We are sorry that we misunderstood that the C δ atom of R^{3.50} was the CZ atom in the PDB file, but it should be the CD atom in the PDB file. Now, we have changed it to the guanidine carbon atom in the figure legend. We also calculated the distance between CZ and CD atoms in R^{6.30} and the CG atom of N^{6.30}, respectively. As shown in Figure R1, the two curves are highly overlapped, indicating that choice of atom in R^{3.50} did not influence the observation of activation tendency in the trajectory.

Fig. R1: Variations of distance from R^{6.30} to N^{6.30} during a representative simulation trajectory. Red and blue lines show CZ and CD as start points in R^{3.50}, respectively.

5. In Figure 4A-C, the weak interactions (such as hbonds) and distances mentioned in the text better to be labeled.

RESPONSE: Thanks for your comments and we have labeled these interactions ($Y^{7.53}-Y^{5.58}$, and $R^{3.50}-N^{6.30}$) with yellow dashed lines and corresponding distances next to them. We have also declared the specific distance value in the revised manuscript (page 15) in order to describe the change during activation directly.

6. In Figure 4E, the area of triangle composed of L3.43, V6.41, and I6.40 is calculated. Please rationalize the chosen of CG atom of L3.43, CB atom of V6.41 and CB atom of I6.40. Why not using CB atoms for all of them?

RESPONSE: Thanks for your suggestion. The atom choice considered the keeping of the original hydrophobic network.

We chose CB of $V^{6.41}$ because $V^{6.41}$ has symmetrical CG1 and CG2. The latter would produce fluctuation in distance measurement but has no influence of the hydrophobic network. As $I^{6.40}$ is next to $V^{6.41}$, we used its CB in the same way as $V^{6.41}$. We chose CG of $L^{3.43}$ because CG is the non-symmetrical atom and closest to the hydrophobic network in $L^{3.43}$. We also added a supplementary information part (Section S10) to state that the choice of the atom did not influence the description of the hydrophobic network. In Fig. S12, we compared the two areas of choosing CG or CB in $L^{3.43}$.

Fig. S12 showed that the choice of CB or CG in $L^{3.43}$ did not impact the tendency of area. As the break of the hydrophobic network is a certain micro-switch in the activation of the AT₁ receptor, the variation in picking atoms for measurement should have no influence on our conclusion.

Fig. S12: Variations of the area of the triangle composed of the C γ (blue line) or C β (red line) atom of L^{3.43}, the C β atom of V^{6.41}, and the C β atom of I^{6.40} in the representative trajectory.

7. How did you design the five clustered mutations in the P6 site? What not just using point mutations?

RESPONSE: We appreciate your comments and have performed single-point mutation experiments in the revised manuscript. Following your suggestions, we designed point mutations with the help of the AlloSigMA server and the result of point mutations also confirmed P6, consistent with our alanine screening experiments.

In the five clustered mutations in our original manuscript, three of them are exactly on P6 and the other two are located at the pathway from P6 to the G protein pocket. The design of P6 site-direct mutations (clusters 1, 4, and 5) is easy to be understood, as the direct perturbation at P6 influences the G protein pocket. Blocking the pathway from P6 to G protein pocket by our mutations (clusters 2 and 3) is anticipated to prove that the signal pathway exists between the two pockets. However, this indirect confirmation is kind of weak, so we deleted this part in the revised manuscript and focused on the direct influence of P6 on G protein binding.

Following your suggestion, we designed point mutations with the help of the AlloSigMA server, which predicts the allosteric perturbation between areas (Guarnera et al., *Bioinformatics* **2017**, 33, 3996–3998; Tan et al., *Nucleic Acids Res.* **2020**, 48, 116–124). AlloSigMA analysis reflected that some P6 residues cause large allosteric

free energy variation in the G protein pocket, indicating that ligand binding influences the fluctuation of G protein pocket, and suggesting bulky or tiny mutations. We also conducted molecular docking using the allosteric GPCR sublibrary of Enamine to further prove important residues for mutation. With the help of guidance, we designed bulky mutations (G^{1.49}L, F^{7.55}W, and F^{8.54}W, Fig. 10A) and tiny mutations (F^{1.48}A, N^{7.49}A, Y^{7.53}A, and F^{8.50}A, Fig. 10B) in the AT₁ receptor and used BRET assays to test the regulation ability of P6 on both G protein and β -arrestin pathways. The β -arrestin 2 activity was severely hampered and Gq activity was also inhibited to a certain degree. Using intramolecular FIAsh-BRET assay, we also confirmed that the conformational ensemble of the AT₁ receptor was changed by the allosteric effect from P6. Thus, we proved the existence of P6 in a relatively complete way.

Our new related part of results was shown below (pages 24-28):

Mutagenesis and FIAsh-BRET assay confirm the predicted cryptic binding site

Next, we confirmed that P6 was a novel site firstly using clustered alanine-scanning mutagenesis. Three groups of mutations (Fig. 9A, B) were independently introduced to directly investigate the effect of allosteric perturbations of P6 on the transducer activity of the AT₁ receptor. The cluster 1 mutations included F^{1.48}A, L^{1.52}A, and I^{1.57}A, located on the intracellular side of TM1. The cluster 2 mutations were N^{7.49}A, P^{7.50}A, and F^{7.55}A on TM7. The cluster 3 mutations (K^{8.49}A, F^{8.50}A, K^{8.51}A, Y^{8.53}A, and F^{8.54}A) are on the top of H8. Since the AT₁ receptor acts as a model system for biased signaling, we further investigated whether these cluster mutations had effects on AngII-induced Gq activation as well as on β -arrestin 2 recruitment using BRET assays upon similar expression (Section 20, SI). Cluster 2 and 3 mutations completely depleted the two activation pathways of the AT₁ receptor, whereas cluster 1 mutations decreased both two E_{max} by 22–34% (Fig. 9C, D, and Section 20, SI). Thus, the direct perturbation at P6 influenced the G protein and β -arrestin pocket.

Fig. 9: (A) The positions of clusters 1 (red), 2 (blue), and 3 (yellow) on the intermediate structure. (B) The sites of alanine scanning clusters on the 7TMs of the AT₁ receptor. (C-D) AngII-induced Gq activation (C) and β-arrestin 2 recruitment (D) through WT AT₁ receptor and three mutants. Three independent experiments were performed, and representative dose-response curves were shown.

To further determine the potential of P6 on our intermediate state to regulate the binding of downstream transducers to the G protein pocket, we applied the AlloSigMA algorithm, which evaluates allosteric effects using Structure-Based the Statistical Mechanical Model of Allostery (Guarnera et al., *Bioinformatics* **2017**, 33, 3996–3998; Tan et al., *Nucleic Acids Res.* **2020**, 48, 116–124). AlloSigMA analysis indicated that ligand binding on P6 influenced the G protein pocket of the receptor and that stable/bulky or unstable/tiny mutations on several P6 residues changed the dynamics of the G protein pocket (Section 21, SI). Thus, we designed bulky mutations (G^{1.49}L, F^{7.55}W, and F^{8.54}W, Fig. 10A) and tiny mutations (F^{1.48}A, N^{7.49}A, Y^{7.53}A, and F^{8.50}A, Fig. 10B) in the AT₁ receptor and used BRET assays to test the regulation ability of P6 on both G protein and β-arrestin pathways. Under similar protein levels (Section 22, SI), point mutations disturbed both Gq and β-arrestin 2 activities (Fig. 10C, 10D, and Section 22, SI). Compared with the Gq signal, β-arrestin 2 signal highly decreased in the presence of the mutations, indicating that P6 is a potential G

protein-biased allosteric site. In all Gq mutations, although the F^{8.54}W mutant retained the same E_{max} as the WT AT₁ receptor, the higher EC₅₀ value suggested a weaker effect of AngII. Thus, point mutations guided by AlloSigMA suggested the existence of P6.

To further investigate the effects of mutations at the potential allosteric site on the AngII-induced AT₁ receptor conformational change, we performed intramolecular FAsH-BRET assays using our recently developed AT₁ receptor conformational sensor, which specifically recognized the Gq-activating conformational state of the receptor (referred to as AT₁ receptor Gq sensor) (Fu et al., *Cell Res.* **2021**, doi:10.1038/s41422-020-00464-8; Li et al., *Nat. Commun.* **2018**, 9, 11). As expected, AngII stimulation on the AT₁ receptor Gq sensor decreased the BRET signal in a dose-dependent manner, suggesting an increase of the distance between the C-terminus and the FAsH motif inserted at the ICL3 of the AT₁ receptor (Fig. 10E, F). Notably, mutations at all seven sites led to the impairment of AngII-induced AT₁ receptor conformational change, as revealed by the increased EC₅₀, whereas G^{1.49}L, F^{1.48}A, N^{7.49}A, Y^{7.53}A, and F^{8.50}A mutants also displayed decreased maximal response (Section 22, SI). These data suggested that the potential allosteric site participated in AT₁ receptor signal regulation through conformational modulation. Hence, considering the shared existence of P6 in the class A GPCR family, it is possible to develop general allosteric modulators targeting P6 and regulating GPCR activation.

Fig. 10: (A-B) The positions of bulky (A) and tiny (B) mutations around P6. (C-D) AngII-induced Gq activation (C) and β-arrestin 2 recruitment (D) in HEK293 cells transfected with WT AT₁ receptor or mutants. (E) AngII-induced conformational change of AT₁ receptor Gq sensor in HEK293 cells transfected with AT₁ receptor Gq sensor or sensor-based mutants. Three independent experiments were performed, and representative dose-response curves were shown in (C-E). The data summary was shown in Section 22, SI.

Reviewers' Comments:

Reviewer #1:

Remarks to the Author:

The revised version of the manuscript presented a large improvement of the work with additional data and suitable changes in the manuscript. I feel that the authors have responded very carefully to my criticism and suggestions. I am indeed happy that the authors found the criticism constructive.

Regarding the added BRET data and sensors, only some minor technical comments and smaller details remain:

1. Fig. 3C-E: The x-axis labeling refers to relative expression levels of the receptor but - if I understand the M&M section correctly - the values on the x-axis represent different amounts of transfected plasmid encoding the respective receptor. Increasing the plasmid amount does not show a 1:1 correlation with increasing the expression levels of the encoded protein and therefore I suggest to change the x-axis labeling to "relative transfected receptor amount".

Since varying luminescence intensity counts can affect the measured BRET signal although the activity state of the biosensor is not altered, the authors should also present the luminescence intensity values of all datapoints presented in Figures 3C-E.

2. Fig. 5: For the assessment of constitutive activity of the receptors in Fig. 5B-D, the authors should also provide the absolute luminescence counts as stated above. Varying luminescence counts can affect the calculated BRET ratio although the activity state of the sensor remains unaltered.

3. Fig. 7: The finding that Y7.53I displays wt-like activity towards Gq but reduced activity towards Gi is very intriguing considering that Gi and Gq are the prime transducers of AT1R. Is this finding in line with recent studies on the structural determinants of GPCR-G protein selectivity?

4. Regarding the statement: "Compared with the Gq signal, β -arrestin 2 signal highly decreased in the presence of the mutations, indicating that P6 is a potential G protein-biased allosteric site" - Isn't it the other way round? Doesn't the finding that P6 mutations affect beta-arrestin coupling more than Gq activation indicate that P6 is an arrestin-biased allosteric site?

5. Fig. 10E: Depicting the sensors used to obtain the data presented in Fig. 10E as Gq sensors (lane 616-620 and 866 in the manuscript) is highly misleading and should be changed. According to the text on page 26 (main file) and the info in the methods section, the sensor that is referred to as "Gq sensor" here is in fact an intramolecular, conformational BRET sensor of the AT1R (Rluc at the CT and a Flash-binding motif in icl3). This biosensor detects conformational changes of AT1R but not Gq dissociation or rearrangement as implicated by the name "Gq sensor". Furthermore, it is not known into which pathway(s) the conformational changes detected by this biosensor are feeding into and it cannot be excluded that BRET signals of this biosensors also reflect activation of pathways other than Gq. The name should be replaced by, for instance, "conformational AT1R sensor" or something along these lines.

Yet, it should of course be mentioned in the main text that the pattern of mutational effects seen with this biosensor rather mirrors the pattern seen in the Gq assay, arguing indeed for the capability of this biosensor to reveal conformational changes primarily - but likely not exclusively - required for AT1R-dependent Gq activation.

Furthermore, where is the comparison of surface and total expression levels of these AT1R sensor mutants (Fig. S23 only shows untagged AT1R)? Differences in BRET changes can quickly be obtained if the sensors are expressed at different total and surface levels in the cells (less at the surface = relatively more BRET from intracellular loci not exposed to agonist treatment  lower delta BRET amplitude).

6. In Fig. S8, the legend should contain the nature of assessment to quantify receptor cell surface expression (most likely ELISA).

Reviewer: Gunnar Schulte

Reviewer #2:

Remarks to the Author:

The manuscript was greatly improved after the revision. It presents an impressive simulation study of the prototypical GPCR AT1R.

We only have a few minor concerns:

- Masculine ordinal indicator 'º' should be replaced by degree sign '°' in lines 190, 196, and 197.
- There should be a space between '>' and '70' in line 212.
- 'G-protein-' should be 'G protein-' in line 215.
- There should be a hyphen '-' following 'β-arrestin' in lines 150 and 215.
- Hyphen '-' should be replaced by dash '-' in lines 326, 407, 497, 595, 632 (twice), and 637.
- 'P2Y1R' should be 'P2Y1 receptor' (to be consistent with other receptor names) in line 561.
- 'β-arrestin-2' should be 'β-arrestin 2' in lines 836 and 837.

Reviewer #1 (Remarks to the Author):

The revised version of the manuscript presented a large improvement of the work with additional data and suitable changes in the manuscript. I feel that the authors have responded very carefully to my criticism and suggestions. I am indeed happy that the authors found the criticism constructive. Regarding the added BRET data and sensors, only some minor technical comments and smaller details remain:

RESPONSE: Thanks for your greatly helpful criticism and suggestions and we are also happy to improve our manuscript under the guidance of an expert in GPCR field. Hope our following responses solve your comments in this time.

1. Fig. 3C-E: The x-axis labeling refers to relative expression levels of the receptor but - if I understand the M&M section correctly - the values on the x-axis represent different amounts of transfected plasmid encoding the respective receptor. Increasing the plasmid amount does not show a 1:1 correlation with increasing the expression levels of the encoded protein and therefore I suggest to change the x-axis labeling to "relative transfected receptor amount".

Since varying luminescence intensity counts can affect the measured BRET signal although the activity state of the biosensor is not altered, the authors should also present the luminescence intensity values of all datapoints presented in Figures 3C-E.

RESPONSE: We agree with the reviewer that the expression levels of the receptors do not show a linear correlation with the transfecting amounts of the plasmids. We indeed carefully controlled the cell surface expression levels of Flag-tagged AT₁R and V₂R in HEK293 cells by adjusting the transfecting amounts of plasmids encoding each receptor. As shown in Fig. S8B, when the relative cell surface expression level of AT₁R in HEK293 cells transfected with 0.1 µg was set as 1, the 2 fold, 4 fold and 8 fold expression of both receptor were achieved by adjusting the transfecting amounts of respective plasmid. Accordingly, the values on the x-axis of Fig 3C-E actually represent different expression levels (1×, 2×, 4×, 8×) of the respective receptor. Thus, we remained the previous x-axis. We apologized for the unclear description in the

original figure legend and has accordingly added the essential information. As for the absolute luminescence intensity values, we have also provided them in the source data file. We also show the corresponding data here in Table R1 and Table R2 for you.

Table R1. The absolute luminescence intensity values for AT₁R in the measurement of constitutive activity for Gq, Gi, and G12.

	RLUC					GFP				
	0	1	2	4	8	0	1	2	4	8
Gq Data1	586393	574071	569485	576607	580522	222126	214645	208716	208789	204402
Gq Data2	572937	587515	566029	563151	577066	215711	217263	206487	201383	201454
Gq Data3	582960	577538	586052	573174	567089	221117	216288	215374	207374	201033
Gi Data1	2672086	2679349	2716731	2675346	2730577	1341654	1338871	1330655	1302091	1303031
Gi Data2	2702887	2739350	2876238	2635347	2799578	1355768	1357348	1405042	1270501	1321961
Gi Data3	2862888	2799351	2734233	2875348	2659579	1436597	1390158	1347430	1388793	1259311
G12 Data1	1335536	1469836	1324421	1583466	1423749	1086191	1180131	1036492	1219427	1060124
G12 Data2	1447580	1489880	1544465	1404610	1488493	1176304	1208740	1222444	1089837	1100294
G12 Data3	1380247	1386547	1441132	1400177	1385460	1123659	1111595	1136621	1076456	1018590

Table R2. The absolute luminescence intensity values for V₂R in the measurement of constitutive activity for Gq, Gi, and G12.

	RLUC					GFP				
	0	1	2	4	8	0	1	2	4	8
Gq Data1	584609	573183	560375	564220	570091	221216	216033	212550	213896	215608
Gq Data2	568430	586944	574065	577981	563852	215151	222041	216652	218881	213982
Gq Data3	567360	575894	562996	566011	572782	215143	217688	213938	215763	217944
Gi Data1	2782348	2619121	2656453	2715266	2870499	1398965	1313751	1333539	1367408	1440129
Gi Data2	2842816	2634272	2816160	2775269	2639500	1431926	1323985	1418781	1394573	1335059
Gi Data3	2702874	2659273	2664155	2765270	2711501	1360627	1341337	1345665	1388442	1360631
G12 Data1	1368091	1374391	1428976	1593021	1373304	1110479	1119854	1158900	1297516	1115398

G12 Data2	1393540	1399840	1454425	1318470	1498753	1132809	1140030	1180557	1069807	1210693
G12 Data3	1349824	1456124	1310709	1574754	1355037	1095517	1179315	1059053	1283425	1104355

Fig. 3: The three macrostates divided by MSM, and constitutive activity. (A) The distribution of three macrostates on the free energy landscape. The attribution and probability of each macrostate are shown on the right. (B) The transition time among the active, inactive, and intermediate states, represented by the mean first passage time. (C-E) Constitutive activities of AT₁ receptor in Gq (C), Gi (D), and G12 (E) pathways. Vasopressin 2 receptor (V₂R) was used as a negative control. The gradient cell surface expression levels of AT₁ receptor and V₂R were achieved by adjusting the transfecting amounts of plasmids encoding the respective receptor in HEK293 cells (Fig S8). Data were from three independent experiments. The bars indicate the mean \pm SEM values. The absolute luminescence intensity values are provided in source data.

2. Fig. 5: For the assessment of constitutive activity of the receptors in Fig. 5B-D, the authors should also provide the absolute luminescence counts as stated above. Varying luminescence counts can affect the calculated BRET ratio although the activity state of the sensor remains unaltered.

RESPONSE: Thanks for your advice and the absolute luminescence intensity values are also provided in corresponding source data. The luminescence counts do not vary between mutants so the measurement is unbiased. We also show the corresponding data here in Table R3 for you.

Table R3. The absolute luminescence intensity values for mutant AT₁R in the measurement of constitutive activity for Gq, Gi, and G12.

	RLUC					GFP				
	0	1	2	4	8	0	1	2	4	8
WT Gq										
Data1	583939	567417	566227	547127	590563	221663	213689	208938	199100	209000
WT Gq										
Data2	589175	561360	593402	528824	578242	221117	206637	215583	188050	202153
WT Gq										
Data3	588251	531795	543445	542290	582528	223477	199210	200205	196309	206215
WT Gi Data1	2605960	2581666	2661432	2690826	2733122	1308974	1283863	1307295	1307472	1302606
WT Gi Data2	2575077	2542326	2727711	2634411	2719145	1297066	1267350	1341215	1274791	1288875
WT Gi Data3	2585215	2656269	2525043	2592715	2609723	1293125	1315650	1234367	1250466	1233616
WT G12										
Data1	1346265	1224775	1352491	1296597	1427883	1092494	984964	1055078	996564	1050494
WT G12										
Data2	1271843	1352631	1401320	1274395	1347020	1034263	1098742	1109425	979373	995717
WT G12										
Data3	1226977	1296795	1259016	1388404	1422808	994710	1045606	987006	1064351	1046475

F ^{6.34} A Gq										
Data1	589085	570685	590748	593261	604009	223086	213094	219463	220337	217322
F ^{6.34} A Gq										
Data2	590079	588364	601004	584119	591939	221162	216577	219427	209874	211085
F ^{6.34} A Gq										
Data3	586501	579472	551157	597152	567489	221287	215042	203184	218378	203473
F ^{6.34} A Gi										
Data1	2629643	2663166	2731593	2807136	2751767	1316399	1318001	1352139	1367356	1323875
F ^{6.34} A Gi										
Data2	2782257	2644323	2808499	2615816	2769284	1396415	1315815	1389926	1279396	1337564
F ^{6.34} A Gi										
Data3	2755519	2658007	2821598	2718848	2611759	1378586	1317042	1394434	1325031	1257040
F ^{6.34} A G12										
Data1	1313697	1402757	1381610	1363637	1326055	1064751	1135111	1104045	1059273	991757
F ^{6.34} A G12										
Data2	1419444	1463161	1344816	1318582	1385300	1152872	1187794	1078273	1023483	1050612
F ^{6.34} A G12										
Data3	1342711	1408546	1315618	1485279	1348002	1089341	1141134	1052626	1152799	1014776
F ^{8.50} A Gq										
Data1	581836	579369	577353	578394	552368	218654	214482	212986	211634	200123
F ^{8.50} A Gq										
Data2	577274	564019	553386	576576	566055	217575	210097	202927	211142	205761
F ^{8.50} A Gq										
Data3	569131	587582	580178	552366	543562	215359	219286	214463	203215	198264
F ^{8.50} A Gi										
Data1	2623956	2833577	2742898	2775392	2664589	1319063	1402337	1356363	1362440	1299254
F ^{8.50} A Gi										
Data2	2624280	2765339	2731675	2599123	2712505	1316339	1367184	1339887	1256936	1311225
F ^{8.50} A Gi										
Data3	2805588	2792176	2865240	2650060	2840867	1410650	1383523	1413566	1293494	1381656

Data3										
F ^{8.50} A G12	1384751	1453418	1330381	1436862	1429987	1124556	1181774	1054061	1136845	1087791
Data1										
F ^{8.50} A G12	1349400	1473434	1409985	1390247	1345984	1098681	1193924	1127424	1095932	1038696
Data2										
F ^{8.50} A G12	1442882	1330093	1419412	1385222	1452713	1169456	1075713	1125594	1089893	1108783
Data3										
K ^{5.42} A Gq										
Data1	559743	526920	537406	581906	572475	212199	198122	200882	217226	212961
K ^{5.42} A Gq										
Data2	539077	594199	597361	540935	561181	203879	225558	223592	201606	208815
K ^{5.42} A Gq										
Data3	598137	547544	554126	546201	581740	225258	205630	206024	202504	215127
K ^{5.42} A Gi										
Data1	2672890	2609353	2746235	2805350	2569581	1343395	1305459	1355816	1381635	1257810
K ^{5.42} A Gi										
Data2	2632721	2812945	2880718	2947422	2659960	1319520	1409286	1438342	1467227	1308168
K ^{5.42} A Gi										
Data3	2667760	2591743	2767579	2750920	2864670	1339482	1299370	1376040	1364044	1407556
K ^{5.42} A G12										
Data1	1346376	1338541	1288570	1411719	1327427	1091776	1085289	1037557	1127681	1024243
K ^{5.42} A G12										
Data2	1328208	1422839	1367052	1361080	1402266	1078771	1142682	1109910	1089681	1094469
K ^{5.42} A G12										
Data3	1453513	1359813	1314398	1478443	1358726	1179236	1096553	1062034	1181498	1053692
W ^{5.62} A Gq										
Data1	573462	561962	571282	581114	567876	216023	210174	210860	213966	207899
W ^{5.62} A Gq										
Data2	589996	592221	553680	575270	567487	221720	220662	201982	209168	204352

W ^{5.62} A Gq										
Data3	571264	582982	564474	593451	571200	215995	218589	208065	218123	208345
W ^{5.62} A Gi										
Data1	2613265	2742084	2650761	2806196	2734031	1311075	1365558	1305235	1382051	1337488
W ^{5.62} AGi										
Data2	2880091	2743046	2699750	2625828	2737499	1449838	1369329	1329357	1289807	1335078
W ^{5.62} A Gi										
Data3	2746379	2611712	2816780	2645895	2640136	1376760	1299457	1384025	1298605	1286802
W ^{5.62} A G12										
Data1	1337662	1451556	1479320	1308445	1423757	1087921	1174744	1183752	1032625	1091167
W ^{5.62} A G12										
Data2	1338175	1466305	1381157	1358566	1304525	1088204	1189613	1102440	1056557	969653
W ^{5.62} A G12										
Data3	1379697	1412546	1351530	1435866	1325420	1121142	1143244	1078859	1123565	999234
Y ^{5.58} A Gq										
Data1	598756	573185	591367	599687	586516	226867	216779	223004	225542	218829
Y ^{5.58} A Gq										
Data2	588887	597458	568837	561179	594420	222835	225242	213378	207868	219282
Y ^{5.58} A Gq										
Data3	563307	564394	570029	578779	571522	213831	213538	214815	216463	212463
Y ^{5.58} A Gi										
Data1	2668391	2760636	2526615	2619928	2848916	1342734	1381146	1259770	1299746	1404231
Y ^{5.58} A Gi										
Data2	2788678	2732007	2843327	2637977	2598598	1398243	1367096	1413987	1303161	1279290
Y ^{5.58} A Gi										
Data3	2523408	2691585	2790855	2759581	2613606	1267255	1347004	1389985	1366407	1287724
Y ^{5.58} A G12										
Data1	1386840	1492848	1454625	1323732	1397058	1127224	1205475	1164718	1058059	1106470
Y ^{5.58} A G12										
Data2	1333668	1443794	1354132	1597897	1344598	1082005	1156046	1083712	1271126	1060350

Fig. 5: (A) The intermediate-specific micro-switches. Involved residues are shown in sticks and corresponding distances are depicted by yellow dashed lines. (B–D) Constitutive and AngII-induced activities of WT AT₁ receptor and mutants in Gq (B), Gi (C), and G12 (D) pathways. Data were from three independent experiments and representative dose-response curves were shown in AngII-induced activity. The summary of data was shown in Section 14, SI. **The absolute luminescence intensity values are provided in source data.**

3. Fig. 7: The finding that Y7.53I displays wt-like activity towards Gq but reduced activity towards Gi is very intriguing considering that Gi and Gq are the prime transducers of AT1R. Is this finding in line with recent studies on the structural determinants of GPCR-G protein selectivity?

RESPONSE: We compared the recently solved class A GPCR-Gq/11 structures with the GPCR-Gi/o complexes. Compared with Gi/o-bound receptors, the features of Gq/11-bound receptors have the increased ICL2 and C-terminal interaction with the G proteins, the extended TM5, and the more outward movement of TM6 (Kim et al., (2020), *Cell*, 182(6), 1574–1588; Maeda et al., (2019), *Science*, 364(6440), 552–557; Xia et al., (2021), *Nat. Commun.*, 12(1), 2086). The more outward TM6 movement in the Gq/11-bound complex than that in the Gi/o-bound complex can be proved in the measurement of activation parameters (68.72° in M₁R with G11, PDB ID: 6OIJ; 66.56° in M₂R with G_{oA}, PDB ID: 6OIK). Since Y^{7.53} has polar interactions with R^{3.50} or Y^{5.58} in macrostates 1 and 7 (Fig. 7A), Y^{7.53}I may quench these interactions and release TM3 and TM5. The flexible TM3 can move downwards and increase the interaction of ICL2 with transducers, while TM5 might form more helices intracellularly. Moreover, Y^{7.53}I weakens the hydrophobic interactions between TM6 and TM7, which possibly promotes the outward movement of TM6 and benefits the recruitment of Gq. According to our hypothesis in Fig. 7, Y^{7.53}I breaks the hydrogen bond with R^{3.50} and inhibits macrostate 1, which is an intermediate state towards the Gi signal. In summary, the effect of Y^{7.53}I on biased signaling is in line with the recently published structures and we have added a corresponding explanation on page 21 in the main text as follows:

The effect of mutations can be confirmed by recent structural information. For instance, Y^{7.53}I quenches polar interactions with R^{3.50} and Y^{5.58}, which releases intracellular TM3 and TM5. The flexible TM3 can move downwards and increase the interaction of ICL2 with transducers, while TM5 might extend towards G protein. Accordingly, the increased ICL2 interaction and extended TM5 are properties of the GPCR-Gq/11 complex compared with GPCR-Gi/o complex^{72–74}. Thus, Y^{7.53}I promotes Gq but inhibits Gi signals in the AT₁ receptor.

4. Regarding the statement: "Compared with the Gq signal, β -arrestin 2 signal highly decreased in the presence of the mutations, indicating that P6 is a potential G protein-biased allosteric site" - Isn't it the other way round? Doesn't the finding that P6 mutations affect beta-arrestin coupling more than Gq activation indicate that P6 is a arrestin-biased allosteric site?

RESPONSE: Thanks for your comments. In the revised manuscript, we did not say the biased potential of the P6 allosteric site. We just stated that P6 is as a novel allosteric site.

5. Fig. 10E: Depicting the sensors used to obtain the data presented in Fig. 10E as Gq sensors (lane 616-620 and 866 in the manuscript) is highly misleading and should be changed. According to the text on page 26 (main file) and the info in the methods section, the sensor that is referred to as "Gq sensor" here is in fact an intramolecular, conformational BRET sensor of the AT1R (Rluc at the CT and a Flash-binding motif in icl3). This biosensor detects conformational changes of AT1R but not Gq dissociation or rearrangement as implicated by the name "Gq sensor". Furthermore, it is not known into which pathway(s) the conformational changes detected by this biosensor are feeding into and it cannot be excluded that BRET signals of this biosensors also reflect activation of pathways other than Gq. The name should be replaced by, for instance, "conformational AT1R sensor" or something along these lines.

Yet, it should of course be mentioned in the main text that the pattern of mutational effects seen with this biosensor rather mirrors the pattern seen in the Gq assay, arguing indeed for the capability of this biosensor to reveal conformational changes primarily - but likely not exclusively - required for AT1R-dependent Gq activation.

Furthermore, where is the comparison of surface and total expression levels of these AT1R sensor mutants (Fig. S23 only shows untagged AT1R)? Differences in BRET

changes can quickly be obtained if the sensors are expressed at different total and surface levels in the cells (less at the surface = relatively more BRET from intracellular loci not exposed to agonist treatment  lower delta BRET amplitude).

RESPONSE: Thanks for your professional suggestions of sensor contents, we will respond to them one by one as follows:

1. We agree with your suggestion and “Gq sensor” in our manuscript has been replaced by “AT₁R conformation sensor” in this revised version.

2. In the revised manuscript, we have stated that the mutational effects shown by the biosensor are consistent with the Gq BRET assay and the biosensors can reflect conformational changes necessary for Gq activation on page 26 as follows:

These mutational effects on the downstream Gq signaling are consistent with the Gq BRET assay, suggesting that the AT₁R conformation sensor can primarily reveal the conformational changes required for the AT₁ receptor Gq signal.

- 3.. The reviewer raised a valuable concern. We indeed measured the total and surface expression levels of the AT₁R conformational sensor as well as sensor-based mutants by using whole-cell ELISA and cell surface ELISA, respectively. As shown in Fig S24, all the mutants showed similar total and cell surface expression levels compared with the AT₁R conformational sensor. This data has been supplemented in our revised manuscript.

Fig. S24: (A-B) Cell surface and total expression levels of AT₁R conformational sensor and sensor-based mutants measured by Cell surface ELISA (A) and whole cell ELISA (B). Data were from three independent experiments. (C-D) Cell surface ELISA (C) and Whole cell ELISA (D) showing equal expression levels of AT₁R conformational sensor and sensor-based mutants. Data were from three independent experiments. n.s., no significant difference; HEK293 cells transfected with AT₁R conformational sensor-based mutants were compared with those transfected with the conformational sensor. The bars indicate the mean \pm SEM values. Statistical differences between WT and mutants were analyzed using one-way ANOVA with Dunnett's post hoc test.

6. In Fig. S8, the legend should contain the nature of assessment to quantify receptor cell surface expression (most likely ELISA).

RESPONSE: Thanks for your carefully reading. The assessment was truly cell surface ELISA and we have provided it in the corresponding figure legend and revised the methodology with more details.

Fig. S8: (A) Cell surface expression levels of AT₁ receptor and vasopressin 2 receptor (V₂R) measured by cell surface ELISA. Data were from three independent experiments. (B) Equal expression levels of AT₁ receptor and V₂R were achieved by adjusting the transfecting amounts in HEK293 cells. Data were from three independent experiments. n.s., no significant difference; HEK293 cells transfected with V₂R were compared with those transfected with AT₁ receptor. The bars indicate the mean ± SEM values. Statistical differences between AT₁ receptor and V₂R were analyzed using unpaired Student's t-test.

Reviewer #2 (Remarks to the Author):

The manuscript was greatly improved after the revision. It presents an impressive simulation study of the prototypical GPCR AT1R.

We only have a few minor concerns:

- Masculine ordinal indicator 'º' should be replaced by degree sign '°' in lines 190, 196, and 197.
- There should be a space between '>' and '70' in line 212.
- 'G-protein-' should be 'G protein-' in line 215.
- There should be a hyphen '-' following 'β-arrestin' in lines 150 and 215.
- Hyphen '-' should be replaced by dash '-' in lines 326, 407, 497, 595, 632 (twice), and 637.
- 'P2Y1R' should be 'P2Y1 receptor' (to be consistent with other receptor names) in line 561.
- 'β-arrestin-2' should be 'β-arrestin 2' in lines 836 and 837.

RESPONSE: We heartfully appreciate the careful reading of the reviewer and these corrections help us a lot to improve our manuscript. According to the reviewer's comments, we have carefully revised the manuscript again and corrected all concerns in the revised manuscript.